# Learnable Graph Convolutional Attention Networks

**Adrián Javaloy** [1*]    **Pablo Sánchez-Martín** [1,2*]    **Amit Levi** [3]    **Isabel Valera** [1,4]

[1]Department of Computer Science of Saarland University, Saarbrücken, Germany
[2]Max Planck Institute for Intelligent Systems, Tübingen, Germany
[3]Huawei Noah's Ark Lab, Montreal, Canada
[4]Max Planck Institute for Software Systems, Saarbrücken, Germany

## Abstract

Existing Graph Neural Networks (GNNs) compute the message exchange between nodes by either *convolving* the features of all the neighboring nodes (GCNs), or by applying *attention* instead (GATs). In this work, we aim at exploiting the strengths of both approaches to their full extent. To this end, we first introduce a graph convolutional attention layer (CAT), which relies on convolutions to compute the attention scores, and theoretically show that there is no clear winner between the three models, as their performance depends on the nature of the data. This brings us to our main contribution, the learnable graph convolutional attention network (L-CAT): a GNN architecture that automatically interpolates between GCN, GAT and CAT in each layer, by introducing two additional (scalar) parameters. Our results demonstrate that L-CAT is able to efficiently combine different GNN layers along the network, outperforming competing methods in a wide range of datasets, and resulting in a more robust model that reduces the need of cross-validating.

## 1   Introduction

Graph Neural Networks (GNNs) [21] have become ubiquitous, emerging as the standard approach in many settings such as topic [22], molecule [11], and link prediction [29]. These applications typically use message-passing GNNs [11] and, depending on how this aggregation is implemented, we can define different types of GNN layers. In this regard, two widely adopted layers are graph convolutional networks (GCNs) [17], which uniformly average the neighboring information; and graph attention networks (GATs) [26], which instead perform a weighted average, based on an attention score between receiver and sender nodes. Recently, a number of works have shown the strengths and limitations of both approaches [3, 4, 10, 18], showing that their performance depends on the data at hand and, thus, computationally demanding cross-validation cannot be avoided. Moreover, the type of layer is usually shared along the GNN, as selecting a model for each layer in the GNN results in a combinatorial number of choices.

In this work, we aim to exploit both convolution and attention operations, while simplifying the design of GNNs. First, we introduce a novel attention layer (CAT), extending attention layers with convolved features. Following [10], we theoretically show that, unfortunately, no free-lunch exists among the three layers, as their performance is data-dependent. As a result, we propose the *learnable graph convolutional attention network* (L-CAT) which, in each layer, automatically interpolates between the three previous operations. L-CAT learns the proper operation to apply, combining them along the GNN architecture, and overcoming the need to cross-validate. Our empirical results show that L-CAT outperforms existing baseline GNNs in terms of both performance and robustness to noise and network initialization.

---

*Equal contribution. Correspondence to: ajavaloy@cs.uni-saarland.de and psanchez@tue.mpg.de.

Has it Trained Yet? Workshop at the Conference on Neural Information Processing Systems (NeurIPS 2022).

## 2 Preliminaries

A message-passing GNN layer yields, for each node $i$, a representation $\tilde{\boldsymbol{h}}_i \in \mathbb{R}^{d'}$, by collecting and aggregating the information from each of its neighbors, and using the aggregated message to update its representation from the previous layer, $\boldsymbol{h}_i \in \mathbb{R}^d$. This can be expressed as follows:

$$\tilde{\boldsymbol{h}}_i = f(\boldsymbol{h}'_i) \quad \text{where} \quad \boldsymbol{h}'_i \stackrel{\text{def}}{=} \sum_{j \in N_i^*} \gamma_{ij} \boldsymbol{W}_v \boldsymbol{h}_j \;, \tag{1}$$

where $N_i^*$ is the neighborhood of node $i$ (including $i$), $\boldsymbol{W}_v \in \mathbb{R}^{d' \times d}$ a learnable matrix, $f$ an elementwise function, and $\gamma_{ij} \in [0,1]$ are coefficients such that $\sum_j \gamma_{ij} = 1$ for each node $i$. Let the input features be $\boldsymbol{h}_i^0$, and the predictions be $\boldsymbol{h}_i^L$, then we can define a message-passing GNN [11] as a sequence of $L$ layers as defined above.

**Graph convolutional networks (GCNs)** [17], in short, simply compute the average of the messages, i.e., they assign the same coefficient $\gamma_{ij} = 1/|N_i^*|$ to every neighbor:

$$\tilde{\boldsymbol{h}}_i = f(\boldsymbol{h}'_i) \quad \text{where} \quad \boldsymbol{h}'_i \stackrel{\text{def}}{=} \frac{1}{|N_i^*|} \sum_{j \in N_i^*} \boldsymbol{W}_v \boldsymbol{h}_j \;, \tag{2}$$

**Graph attention networks**, instead of assigning a fixed value to each coefficient $\gamma_{ij}$, compute it as a function of the sender and receiver nodes. Mathematically, it can be written as follows:

$$\tilde{\boldsymbol{h}}_i = f(\boldsymbol{h}'_i) \quad \text{where} \quad \boldsymbol{h}'_i \stackrel{\text{def}}{=} \sum_{j \in N_i^*} \gamma_{ij} \boldsymbol{W}_v \boldsymbol{h}_j \quad \text{and} \quad \gamma_{ij} \stackrel{\text{def}}{=} \frac{\exp(\Psi(\boldsymbol{h}_i, \boldsymbol{h}_j))}{\sum_{\ell \in N_i^*} \exp(\Psi(\boldsymbol{h}_i, \boldsymbol{h}_\ell))} \;. \tag{3}$$

Here, $\Psi(\boldsymbol{h}_i, \boldsymbol{h}_j) \stackrel{\text{def}}{=} \alpha(\boldsymbol{W}_q \boldsymbol{h}_i, \boldsymbol{W}_k \boldsymbol{h}_j)$ is known as the *score function*, and it provides a score value given two feature nodes. The (attention) coefficients are obtained by normalizing these scores. We can find in the literature different attention layers. In this work, we focus on the original GAT [26], and its extension GATv2 [7], and assume that $\boldsymbol{W}_q = \boldsymbol{W}_k = \boldsymbol{W}_v$.

## 3 On the limitations of GCN and GAT networks

Baranwal et al. [3] showed that, when the graph is neither too sparse nor noisy, applying one layer of graph convolution increases the regime in which the data is linearly separable (and thus easier to classify). However, this result is highly sensitive to the graph structure, as convolutions essentially collapse the data to the same value in the presence of enough noise. More recently, Fountoulakis et al. [10] showed that GAT remedies this issue and provides perfect node separability regardless of the noise level in the graph. However, a classical argument (see [1]) states that, *in their particular setting*, a linear classifier already achieves perfect separability, casting graph-based models as unnecessary. These works, in summary, showed scenarios for which GCNs can be beneficial in the absence of noise, and that GATs can outperform GCNs in some scenarios, leaving open the question of which architecture (GCN or GAT) is preferable during deployment.

## 4 Convolved attention: benefits and hurdles

In this section, we propose to combine attention with convolution operations. To motivate it, we complement the results of [10], providing a synthetic dataset for which *any* 1-layer GCN fails, but 1-layer GAT does not. Thus, proving a clear distinction between GAT and GCN layers. Besides, we show that convolution helps GAT as long as the graph noise is reasonable. The proofs for the two statements in this section appear in Appendix A and follow similar arguments as in [10].

This dataset is based on the *contextual stochastic block model* (CSBM) [8]. Let $\varepsilon_1, \ldots, \varepsilon_n$ be i.i.d. uniform samples from $\{-1, 0, 1\}$. Let $C_k = \{j \in [n] \mid \varepsilon_j = k\}$ for $k \in \{-1, 0, 1\}$. We set the feature vector $\mathbf{X}_i \sim \mathcal{N}(\varepsilon_i \boldsymbol{\mu}, \mathbf{I}\sigma^2)$ where $\boldsymbol{\mu} \in \mathbb{R}^d$, $\sigma \in \mathbb{R}$. For a given pair $p, q \in [0,1]$ we consider the stochastic adjacency matrix $\mathbf{A} \in \{0,1\}^{n \times n}$ defined as follows: for $i, j \in [n]$ in the same class (*intra-edge*), we set $a_{ij} \sim \text{Ber}(p)$; for $i, j$ in different classes (*inter-edge*), we set $a_{ij} \sim \text{Ber}(q)$.

We denote by $(\mathbf{X}, \mathbf{A}) \sim \mathsf{CSBM}(n, p, q, \boldsymbol{\mu}, \sigma^2)$ a sample obtained according to the above random process. Our task is then to distinguish (or separate) nodes from $C_0$ vs. $C_{-1} \cup C_1$.

In general, it is impossible to separate $C_0$ from $C_{-1} \cup C_1$ with a linear classifier; and using a convolutional layer is detrimental for node classification:[2] although the convolution brings the means closer and shrinks the variance, the geometric structure of the problem does not change. In contrast, GAT can achieve perfect node separability if the graph is not too sparse:

**Theorem 1.** Suppose that $p, q = \Omega(\log^2 n/n)$ and $\|\boldsymbol{\mu}\|_2 = \omega(\sigma\sqrt{\log n})$. Then, there exists a choice of attention architecture $\Psi$ such that, with probability at least $1 - o_n(1)$ over the data $(\mathbf{X}, \mathbf{A}) \sim \mathsf{CSBM}(n, p, q, \boldsymbol{\mu}, \sigma^2)$, GAT separates nodes $C_0$ from $C_1 \cup C_{-1}$.

Moreover, we show that the above threshold $\|\boldsymbol{\mu}\|$ can be improved when the graph noise is reasonable. Specifically, *by applying convolution prior to the attention score*, the variance of the data is greatly reduced and, if the graph is not too noisy, this operation dramatically lowers the bound on $\|\boldsymbol{\mu}\|$. We exploit this insight by introducing the *graph convolutional attention layer* (CAT):

$$\Psi(\boldsymbol{h}_i, \boldsymbol{h}_j) = \alpha(\boldsymbol{W}\tilde{\boldsymbol{h}}_i, \boldsymbol{W}\tilde{\boldsymbol{h}}_j) \quad \text{where} \quad \tilde{\boldsymbol{h}}_i = \frac{1}{|N_i^*|}\sum_{\ell \in N_i^*} \boldsymbol{h}_\ell \,, \tag{4}$$

where $\tilde{\boldsymbol{h}}_i$ are the convolved features from the neighborhood of node $i$. As we show now, CAT improves over GAT by combining convolutions with attention, when the graph noise is low.

**Corollary 2.** Suppose $p, q = \Omega(\log^2 n/n)$ and $\|\boldsymbol{\mu}\| \geq \omega\left(\sigma\sqrt{\frac{(p+2q)\log n}{n(p-q)^2}}\right)$. Then, there is a choice of attention architecture $\Psi$ such that CAT separates nodes $C_0$ from $C_1 \cup C_{-1}$ with probability at least $1 - o(1)$ over the data $(\mathbf{X}, \mathbf{A}) \sim \mathsf{CSBM}(n, p, q, \boldsymbol{\mu}, \sigma^2)$.

## 5  L-CAT: Learning to interpolate

From the previous analysis, we conclude that it is hard to know *a priori* whether attention, convolution, or convolved attention, will perform the best. We argue that this issue can be easily overcome by learning to interpolate between the three. To this end, we propose the *learnable convolutional attention layer* (L-CAT), which can be formulated as an attention layer as follows:

$$\Psi(\boldsymbol{h}_i, \boldsymbol{h}_j) = \lambda_1 \cdot \alpha(\boldsymbol{W}\tilde{\boldsymbol{h}}_i, \boldsymbol{W}\tilde{\boldsymbol{h}}_j) \quad \text{where} \quad \tilde{\boldsymbol{h}}_i = \frac{\boldsymbol{h}_i + \lambda_2 \sum_{\ell \in N_i} \boldsymbol{h}_\ell}{1 + \lambda_2|N_i|} \,, \tag{5}$$

where $\lambda_1, \lambda_2 \in [0, 1]$. Here, $\lambda_1$ interpolates between pure attention scores and convolutions, while $\lambda_2$ adjusts the amount of convolution within the score function. In short, L-CAT learns to interpolate between GCN ($\lambda_1 = 0$), GAT ($\lambda_1 = 1$ and $\lambda_2 = 0$), and CAT ($\lambda_1 = 1$ and $\lambda_2 = 1$).

Despite its simplicity, L-CAT provides a number of non-trivial benefits. Not only can it switch between layer types, but it also learns the amount of attention necessary for each use-case. Moreover, by comprising the three layers in a single learnable formulation, it removes the necessity of cross-validating the type of layer, as their performance is data-dependent (see §§3 and 4). Remarkably, it easily allows us to combine different layer types within the same architecture.

## 6  Experiments

**Synthetic data**  First, we empirically validate our theoretical results (Thm. 1 and Cor. 2). We aim to understand the behavior of each layer as the properties of the data change, i.e., the noise level $q$ (proportion of inter-edges) and the distance between the means of consecutive classes $\|\boldsymbol{\mu}\|$. We provide in App. B the experimental setup, extra results and experiments. Figure 1 (left) shows performance on the hard and easy regimes, as we vary the noise level $q$. In the hard regime, GAT cannot separate for any value of $q$, whereas CAT perfectly classifies for small $q$. When the means are far apart, GAT achieves perfect results for any $q$, as stated in Thm. 1. In contrast, when CAT fails to satisfy the condition in Cor. 2 it achieves inferior performance. In the right-most part of Fig. 1, we fix

---
[2]We note that this problem can be easily solved by two layers of GCN [4].

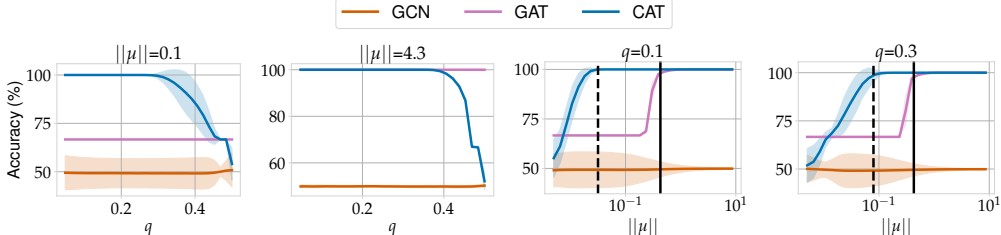

Figure 1: Synthetic data results. In the two left-most figures, we show how the accuracy varies with the noise level $q$ for $\|\boldsymbol{\mu}\| = 0.1$ and $\|\boldsymbol{\mu}\| = 4.3$. In the two right-most figures, we show how the accuracy varies with the norm of the means $\|\boldsymbol{\mu}\|$ for $q = 0.1$ and $q = 0.3$. We use two vertical lines to present the classification threshold stated in Thm. 1 (solid line) and Cor. 2 (dashed line).

Table 1: Test accuracy (%), and averaged over ten runs. Bold numbers are statistically different to their baseline model ($\alpha = 0.05$). Best average performance is underlined.

| Dataset | Amazon Computers | Amazon Photo | GitHub | Facebook PagePage | Coauthor Physics | TwitchEN |
|---|---|---|---|---|---|---|
| Avg. Deg. | 35.76 | 31.13 | 15.33 | 15.22 | 14.38 | 10.91 |
| GCN | $\underline{90.59 \pm 0.36}$ | $\underline{95.13 \pm 0.57}$ | $\underline{84.13 \pm 0.44}$ | $94.76 \pm 0.19$ | $96.36 \pm 0.10$ | $57.83 \pm 1.13$ |
| GAT | $89.59 \pm 0.61$ | $94.02 \pm 0.66$ | $83.31 \pm 0.18$ | $94.16 \pm 0.48$ | $96.36 \pm 0.10$ | $57.59 \pm 1.20$ |
| CAT | $\mathbf{90.58 \pm 0.40}$ | $\mathbf{94.77 \pm 0.47}$ | $\mathbf{84.11 \pm 0.66}$ | $\mathbf{94.71 \pm 0.30}$ | $96.40 \pm 0.10$ | $\underline{58.09 \pm 1.61}$ |
| L-CAT | $\mathbf{90.34 \pm 0.47}$ | $\mathbf{94.93 \pm 0.37}$ | $84.05 \pm 0.70$ | $\underline{\mathbf{94.81 \pm 0.25}}$ | $96.35 \pm 0.10$ | $57.88 \pm 2.07$ |

$q$ and sweep $\|\boldsymbol{\mu}\|$, clearly showing the transition in accuracy of both GAT and CAT as a function of $\|\boldsymbol{\mu}\|$. As we increase $q$, the improvement of CAT over GAT decreases, as stated in Cor. 2.

**Real data** We study GCN, GAT, CAT and L-CAT, in six node classification datasets. See Appendices C and D. In Table 1, we find GCN to be a strong contender, reinforcing its viability in real-world data despite its simplicity. CAT and L-CAT not only hold up, but they mostly improve test accuracy with respect to the attention baseline model. Moreover, we find a positive correlation between performance improvement of CAT and L-CAT (with respect to GAT), and the average node degree of the graph, as shown in the inset figure.

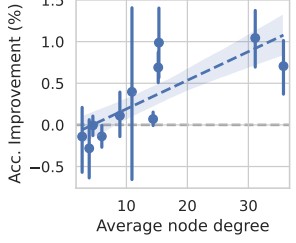

**Robustness to noise** We consider the *arxiv* dataset from the OGB suite [13], as it is more difficult to solve and tune. Refer to App. E for further details and results. We add additive noise to the node features of the form $\mathcal{N}(\mathbf{0}, \mathbf{1}\sigma)$, and consider different noise levels. The inset figure shows that GAT models are sensitive to noise and exhibit high variance. GCNs are more robust to noise and have almost no variance. As expected, CAT reduces the variance by leveraging convolutions, and L-CAT is the most robust model, providing better performance and reduced variance.

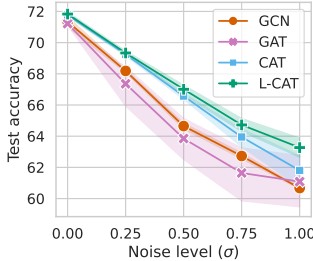

**Layer robustness** To assess the consistency of the $\lambda$ parameters, the inset figure shows their evolution while training on the *arxiv* dataset. We observe that L-CATv2 converged to a GNN that combines three types of layers: i) a CATv2 layer, taking advantage of the neighboring information; ii) a quasi-GCN layer, where scores are almost uniform and some neighboring information is still used in the score; and iii) a pure GCN layer, with uniform scores.

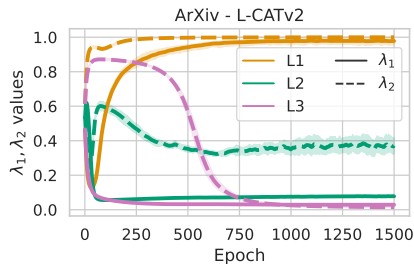

# 7 Conclusions

In this work, we studied how to combine the strengths of convolution and attention layers in GNNs. We proposed CAT, which computes attention with respect to the convolved features, and analyzed its benefits and limitations on a new synthetic dataset. This analysis revealed different regimes where one model is preferred over the others, reinforcing the idea that selecting between GCNs, GATs, and now CATs, is a difficult task, as their performance directly depend on the data. For this reason, we proposed L-CAT, a model which is able to interpolate between the three via two learnable parameters. Extensive experimental results demonstrated the effectiveness of L-CAT, yielding great results while being more robust than other methods due to its adaptability. As a result, L-CAT proved to be a viable drop-in replacement that removes the need to cross-validate the layer type.

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

# Appendix

## Table of Contents

# A Theoretical results

## A.1 A hard example for GCN

In this subsection, we present a dataset and classification task for which GCN performs poorly. Note that we follow the similar techniques and notation as [10], as described in the main paper.

We recall our data model. Fix $n, d \in \mathbb{N}$ and let $\varepsilon_1, \ldots, \varepsilon_n$ be i.i.d uniformly sampled from $\{-1, 0, 1\}$. Let $C_k = \{j \in [n] \mid \varepsilon_j = k\}$ for $k \in \{-1, 0, 1\}$. For each index $i \in [n]$, we set the feature vector $\mathbf{X}_i \in \mathbb{R}^d$ as $\mathbf{X}_i \sim \mathcal{N}(\varepsilon_i \cdot \boldsymbol{\mu}, \mathbf{I} \cdot \sigma^2)$, where $\boldsymbol{\mu} \in \mathbb{R}^d$, $\sigma \in \mathbb{R}$ and $\mathbf{I} \in \{0, 1\}^{d \times d}$ is the identity matrix. For a given pair $p, q \in [0, 1]$ we consider the stochastic adjacency matrix $\mathbf{A} \in \{0, 1\}^{n \times n}$ defined as follows. For $i, j \in [n]$ in the same class, we set $a_{ij} \sim \mathrm{Ber}(p)$, and if $i, j$ are in different classes, we set $a_{ij} \sim \mathrm{Ber}(q)$. We let $\mathbf{D} \in \mathbb{R}^{n \times n}$ be a diagonal matrix containing the degrees of the vertices. We denote by $(\mathbf{X}, \mathbf{A}) \sim \mathsf{CSBM}(n, p, q, \boldsymbol{\mu}, \sigma^2)$ a sample obtained according to the above random process.

The task we wish to solve is classifying $C_0$ vs $C_{-1} \cup C_1$. Namely, we want our model $\varphi$ to satisfy $\varphi(\mathbf{X}_i) < 0$ if and only if $i \in C_0$. Moreover, note that the posed problem *is not linearly classifiable*.

To this end, we start by stating an assumption on the choice of parameters. This assumption is necessary to achieve degree concentration in the graph.

**Assumption 1.** $p, q = \Omega(\log^2 n / n)$ .

We now show the distribution of the convolved features. The following lemma can be easily obtained using the techniques in [3].

**Lemma 3.** Fix $p, q$ satisfying Assumption 1. With probability at least $1 - o(1)$ over $\mathbf{A}$ and $\{\varepsilon_i\}_i$,

$$(\mathbf{D}^{-1}\mathbf{A}\mathbf{X})_i \sim \mathcal{N}\left(\varepsilon_i \cdot \frac{p - q}{p + 2q}\boldsymbol{\mu}, \frac{\sigma^2}{n(p + 2q)}\right), \qquad \forall i \in [n].$$

To prove the above lemma, we need the following definition of our high probability event.

**Definition 1.** We define the even $\mathcal{E}$ as the intersection of the following events over $\mathbf{A}$ and $\{\varepsilon_i\}_i$:

1. $\mathcal{E}_1$ is the event that $|C_0| = \frac{n}{3} \pm O(\sqrt{n \log n})$, $|C_1| = \frac{n}{3} \pm O(\sqrt{n \log n})$ and $|C_{-1}| = \frac{n}{3} \pm O(\sqrt{n \log n})$.

2. $\mathcal{E}_2$ is the event that for each $i \in [n]$, $\mathbf{D}_{ii} = \frac{n(p+2q)}{3}\left(1 \pm \frac{10}{\sqrt{\log n}}\right)$.

3. $\mathcal{E}_3$ is the event that for each $i \in [n]$ and $k \in \{-1, 0, 1\}$,

$$|N_i \cap C_k| = \begin{cases} \mathbf{D}_{ii} \cdot \frac{p}{p+2q} \cdot \left(1 \pm \frac{10}{\sqrt{\log n}}\right) & \text{if } i \in C_k \\ \mathbf{D}_{ii} \cdot \frac{q}{p+2q} \cdot \left(1 \pm \frac{10}{\sqrt{\log n}}\right) & \text{if } i \notin C_k \end{cases}.$$

The following lemma is a direct application of Chernoff bound and a union bound.

**Lemma 4.** With probability at least $1 - 1/\mathrm{poly}(n)$ the event $\mathcal{E}$ holds.

**Proof of Lemma 3.** By applying Lemma 4, and conditioned on $\mathcal{E}$, for any $i \in [n]$

$$(\mathbf{D}^{-1}\mathbf{A}\mathbf{X})_i = \frac{1}{\mathbf{D}_{ii}} \sum_{j \in N_i} \mathbf{X}_j = \frac{1}{\mathbf{D}_{ii}} \left( \sum_{j \in N_i \cap C_{-1}} \mathbf{X}_j + \sum_{j \in N_i \cap C_0} \mathbf{X}_j + \sum_{j \in N_i \cap C_1} \mathbf{X}_j \right).$$

Using the definition of $\mathcal{E}$ and properties of Gaussian distributions the lemma follows. $\qquad \square$

Lemma 3 shows that essentially, the convolution reduced the variance and moved the means closer, but the structure of the problem stayed exactly the same. Therefore, one layer of GCN cannot separate $C_0$ from $C_{-1} \cup C_1$ with high probability.

## A.2 A solution for GAT and CAT

In what follows, we show that GAT is able to handle the above classification task easily when the distance between the means is large enough. Then, we show how the additional convolution on the inputs to the score function improves the regime of perfect classification when the graph is not too noisy. Our main technical lemma considers a specific attention architecture and characterize the attention scores for our data model.

**Lemma 5.** Suppose that $p, q$ satisfy Assumption 1, $\|\boldsymbol{\mu}\| \geq \omega\left(\sigma\sqrt{\log n}\right)$, fix the LeakyRelu constant $\beta \in (0, 1)$ and $R \in \mathbb{R}$. Then, there exists a choice of attention architecture $\Psi$ such that with probability at least $1 - o_n(1)$ over the data $(\mathbf{X}, \mathbf{A}) \sim \mathsf{CSBM}(n, p, q, \boldsymbol{\mu}, \sigma^2)$ the following holds.

$$
\Psi(\mathbf{X}_i, \mathbf{X}_j) = \begin{cases}
10R\beta\|\boldsymbol{\mu}\|(1 \pm o(1)) & \text{if } i, j \in C_1^2 \\
-2R\|\boldsymbol{\mu}\|(1 + 2\beta)(1 \pm o(1)) & \text{if } i, j \in C_{-1}^2 \\
-2R\|\boldsymbol{\mu}\|(1 + 5\beta)(1 \pm o(1)) & \text{if } i \in C_1, \ j \in C_{-1} \\
10R\beta\|\boldsymbol{\mu}\|(1 \pm o(1)) & \text{if } i \in C_{-1}, \ j \in C_1 \\
-\frac{R}{2}\|\boldsymbol{\mu}\|(1 - 21\beta)(1 \pm o(1)) & \text{if } i \in C_0, \ j \in C_1 \\
-\frac{R}{2}\|\boldsymbol{\mu}\|(1 - 11\beta)(1 \pm o(1)) & \text{if } i \in C_0, \ j \in C_{-1} \\
-\frac{R}{2}\|\boldsymbol{\mu}\|(1 - 5\beta)(1 \pm o(1)) & \text{if } i \in C_1, \ j \in C_0 \\
-\frac{R}{2}\|\boldsymbol{\mu}\|(1 - 5\beta)(1 \pm o(1)) & \text{if } i \in C_{-1}, \ j \in C_0 \\
2R\beta\|\boldsymbol{\mu}\|(1 \pm o(1)) & \text{if } i, j \in C_0^2
\end{cases}.
$$

**Proof.** We consider as an ansatz the following two layer architecture $\Psi$.

$$
\tilde{\boldsymbol{w}} \stackrel{\text{def}}{=} \frac{\boldsymbol{\mu}}{\|\boldsymbol{\mu}\|}, \qquad
\mathbf{S} \stackrel{\text{def}}{=} \begin{bmatrix} 1 & 1 \\ -1 & -1 \\ 1 & -1 \\ -1 & 1 \\ 0 & 1 \\ 1 & 0 \\ 0 & -1 \\ -1 & 0 \end{bmatrix}, \qquad
\boldsymbol{b} \stackrel{\text{def}}{=} \begin{bmatrix} -3/2 \\ -3/2 \\ -3/2 \\ -3/2 \\ -1/2 \\ -1/2 \\ -1/2 \\ -1/2 \end{bmatrix} \cdot \|\boldsymbol{\mu}\|, \qquad
\boldsymbol{r} \stackrel{\text{def}}{=} R \cdot \begin{bmatrix} 2 \\ -2 \\ -2 \\ 2 \\ -1 \\ -1 \\ -1 \\ -1 \end{bmatrix},
$$

where $R > 0$ is an arbitrary scaling parameter. The output of the attention model is defined as

$$
\Psi(\mathbf{X}_i, \mathbf{X}_j) \stackrel{\text{def}}{=} \boldsymbol{r}^T \cdot \mathrm{LeakyRelu}\left(\mathbf{S} \cdot \begin{bmatrix} \tilde{\boldsymbol{w}}^T\mathbf{X}_i \\ \tilde{\boldsymbol{w}}^T\mathbf{X}_j \end{bmatrix} + \boldsymbol{b}\right).
$$

Let $\boldsymbol{\Delta}_{ij} \stackrel{\text{def}}{=} \mathbf{S} \cdot \begin{bmatrix} \tilde{\boldsymbol{w}}^T\mathbf{X}_i \\ \tilde{\boldsymbol{w}}^T\mathbf{X}_j \end{bmatrix} + \boldsymbol{b} \in \mathbb{R}^8$, and note that for each element $t \in [8]$ of $\boldsymbol{\Delta}_{ij}$, we have that $(\boldsymbol{\Delta}_{ij})_t = \mathbf{S}_{t,1}\tilde{\boldsymbol{w}}^T\mathbf{X}_i + \mathbf{S}_{t,2}\tilde{\boldsymbol{w}}^T\mathbf{X}_j + \boldsymbol{b}_t$. Note that the random variable $(\boldsymbol{\Delta}_{ij})_t$ is distributed as follows:

$$
(\boldsymbol{\Delta}_{ij})_t \sim \begin{cases}
\mathcal{N}\left((\mathbf{S}_{t,1} + \mathbf{S}_{t,2})\tilde{\boldsymbol{w}}^T\boldsymbol{\mu} + \boldsymbol{b}_t, \ \|\mathbf{S}_{t,*}\|^2\sigma^2\right) & \text{if } i, j \in C_1^2 \\
\mathcal{N}\left(-(\mathbf{S}_{t,1} + \mathbf{S}_{t,2})\tilde{\boldsymbol{w}}^T\boldsymbol{\mu} + \boldsymbol{b}_t, \ \|\mathbf{S}_{t,*}\|^2\sigma^2\right) & \text{if } i, j \in C_{-1}^2 \\
\mathcal{N}\left((\mathbf{S}_{t,1} - \mathbf{S}_{t,2})\tilde{\boldsymbol{w}}^T\boldsymbol{\mu} + \boldsymbol{b}_t, \ \|\mathbf{S}_{t,*}\|^2\sigma^2\right) & \text{if } i \in C_1, \ j \in C_{-1} \\
\mathcal{N}\left(-(\mathbf{S}_{t,1} - \mathbf{S}_{t,2})\tilde{\boldsymbol{w}}^T\boldsymbol{\mu} + \boldsymbol{b}_t, \ \|\mathbf{S}_{t,*}\|^2\sigma^2\right) & \text{if } i \in C_{-1}, \ j \in C_1 \\
\mathcal{N}\left(\mathbf{S}_{t,2}\tilde{\boldsymbol{w}}^T\boldsymbol{\mu} + \boldsymbol{b}_t, \ \|\mathbf{S}_{t,*}\|^2\sigma^2\right) & \text{if } i \in C_0, \ j \in C_1 \\
\mathcal{N}\left(-\mathbf{S}_{t,2}\tilde{\boldsymbol{w}}^T\boldsymbol{\mu} + \boldsymbol{b}_t, \ \|\mathbf{S}_{t,*}\|^2\sigma^2\right) & \text{if } i \in C_0, \ j \in C_{-1} \\
\mathcal{N}\left(\mathbf{S}_{t,1}\tilde{\boldsymbol{w}}^T\boldsymbol{\mu} + \boldsymbol{b}_t, \ \|\mathbf{S}_{t,*}\|^2\sigma^2\right) & \text{if } i \in C_1, \ j \in C_0 \\
\mathcal{N}\left(-\mathbf{S}_{t,1}\tilde{\boldsymbol{w}}^T\boldsymbol{\mu} + \boldsymbol{b}_t, \ \|\mathbf{S}_{t,*}\|^2\sigma^2\right) & \text{if } i \in C_{-1}, \ j \in C_0 \\
\mathcal{N}\left(\boldsymbol{b}_t, \ \|\mathbf{S}_{t,*}\|^2\sigma^2\right) & \text{if } i, j \in C_0^2
\end{cases}.
$$

Therefore, for a fixed $i, j \in [n]^2$ we have that the entries of $\boldsymbol{\Delta}_{ij}$ are distributed as follows (where we use $\mathcal{N}_x^y$ as abbreviation for the Gaussian $\mathcal{N}(x, y)$)

$$
\begin{bmatrix} \mathcal{N}_{\frac{\|\boldsymbol{\mu}\|}{2}}^{4\sigma^2} & \mathcal{N}_{\frac{-7\|\boldsymbol{\mu}\|}{2}}^{4\sigma^2} & \mathcal{N}_{-\frac{3\|\boldsymbol{\mu}\|}{2}}^{4\sigma^2} & \mathcal{N}_{-\frac{3\|\boldsymbol{\mu}\|}{2}}^{4\sigma^2} & \mathcal{N}_{\frac{\|\boldsymbol{\mu}\|}{2}}^{\sigma^2} & \mathcal{N}_{\frac{\|\boldsymbol{\mu}\|}{2}}^{\sigma^2} & \mathcal{N}_{-\frac{3\|\boldsymbol{\mu}\|}{2}}^{\sigma^2} & \mathcal{N}_{-\frac{3\|\boldsymbol{\mu}\|}{2}}^{\sigma^2} \end{bmatrix} \qquad \text{for } i, j \in C_1^2,
$$

$$\left[\mathcal{N}^{4\sigma^2}_{-\frac{7\|\boldsymbol{\mu}\|}{2}} \quad \mathcal{N}^{4\sigma^2}_{\frac{\|\boldsymbol{\mu}\|}{2}} \quad \mathcal{N}^{4\sigma^2}_{-\frac{3\|\boldsymbol{\mu}\|}{2}} \quad \mathcal{N}^{4\sigma^2}_{-\frac{3\|\boldsymbol{\mu}\|}{2}} \quad \mathcal{N}^{\sigma^2}_{-\frac{3\|\boldsymbol{\mu}\|}{2}} \quad \mathcal{N}^{\sigma^2}_{-\frac{3\|\boldsymbol{\mu}\|}{2}} \quad \mathcal{N}^{\sigma^2}_{\frac{\|\boldsymbol{\mu}\|}{2}} \quad \mathcal{N}^{\sigma^2}_{\frac{\|\boldsymbol{\mu}\|}{2}}\right] \qquad \text{for } i,j \in C^2_{-1},$$

$$\left[\mathcal{N}^{4\sigma^2}_{-\frac{3\|\boldsymbol{\mu}\|}{2}} \quad \mathcal{N}^{4\sigma^2}_{-\frac{3\|\boldsymbol{\mu}\|}{2}} \quad \mathcal{N}^{4\sigma^2}_{\frac{\|\boldsymbol{\mu}\|}{2}} \quad \mathcal{N}^{4\sigma^2}_{-\frac{7\|\boldsymbol{\mu}\|}{2}} \quad \mathcal{N}^{\sigma^2}_{-\frac{3\|\boldsymbol{\mu}\|}{2}} \quad \mathcal{N}^{\sigma^2}_{\frac{\|\boldsymbol{\mu}\|}{2}} \quad \mathcal{N}^{\sigma^2}_{\frac{\|\boldsymbol{\mu}\|}{2}} \quad \mathcal{N}^{\sigma^2}_{-\frac{3\|\boldsymbol{\mu}\|}{2}}\right] \qquad \text{for } i,j \in C_1 \times C_{-1},$$

$$\left[\mathcal{N}^{4\sigma^2}_{-\frac{3\|\boldsymbol{\mu}\|}{2}} \quad \mathcal{N}^{4\sigma^2}_{-\frac{3\|\boldsymbol{\mu}\|}{2}} \quad \mathcal{N}^{4\sigma^2}_{-\frac{7\|\boldsymbol{\mu}\|}{2}} \quad \mathcal{N}^{4\sigma^2}_{\frac{\|\boldsymbol{\mu}\|}{2}} \quad \mathcal{N}^{\sigma^2}_{\frac{\|\boldsymbol{\mu}\|}{2}} \quad \mathcal{N}^{\sigma^2}_{-\frac{3\|\boldsymbol{\mu}\|}{2}} \quad \mathcal{N}^{\sigma^2}_{-\frac{3\|\boldsymbol{\mu}\|}{2}} \quad \mathcal{N}^{\sigma^2}_{\frac{\|\boldsymbol{\mu}\|}{2}}\right] \qquad \text{for } i,j \in C_{-1} \times C_1,$$

$$\left[\mathcal{N}^{4\sigma^2}_{-\frac{\|\boldsymbol{\mu}\|}{2}} \quad \mathcal{N}^{4\sigma^2}_{-\frac{5\|\boldsymbol{\mu}\|}{2}} \quad \mathcal{N}^{4\sigma^2}_{-\frac{5\|\boldsymbol{\mu}\|}{2}} \quad \mathcal{N}^{4\sigma^2}_{-\frac{\|\boldsymbol{\mu}\|}{2}} \quad \mathcal{N}^{\sigma^2}_{\frac{\|\boldsymbol{\mu}\|}{2}} \quad \mathcal{N}^{\sigma^2}_{-\frac{\|\boldsymbol{\mu}\|}{2}} \quad \mathcal{N}^{\sigma^2}_{-\frac{3\|\boldsymbol{\mu}\|}{2}} \quad \mathcal{N}^{\sigma^2}_{-\frac{\|\boldsymbol{\mu}\|}{2}}\right] \qquad \text{for } i,j \in C_0 \times C_1,$$

$$\left[\mathcal{N}^{4\sigma^2}_{-\frac{5\|\boldsymbol{\mu}\|}{2}} \quad \mathcal{N}^{4\sigma^2}_{-\frac{\|\boldsymbol{\mu}\|}{2}} \quad \mathcal{N}^{4\sigma^2}_{-\frac{\|\boldsymbol{\mu}\|}{2}} \quad \mathcal{N}^{4\sigma^2}_{-\frac{5\|\boldsymbol{\mu}\|}{2}} \quad \mathcal{N}^{\sigma^2}_{-\frac{3\|\boldsymbol{\mu}\|}{2}} \quad \mathcal{N}^{\sigma^2}_{-\frac{\|\boldsymbol{\mu}\|}{2}} \quad \mathcal{N}^{\sigma^2}_{\frac{\|\boldsymbol{\mu}\|}{2}} \quad \mathcal{N}^{\sigma^2}_{-\frac{\|\boldsymbol{\mu}\|}{2}}\right] \qquad \text{for } i,j \in C_0 \times C_{-1},$$

$$\left[\mathcal{N}^{4\sigma^2}_{-\frac{\|\boldsymbol{\mu}\|}{2}} \quad \mathcal{N}^{4\sigma^2}_{-\frac{5\|\boldsymbol{\mu}\|}{2}} \quad \mathcal{N}^{4\sigma^2}_{-\frac{\|\boldsymbol{\mu}\|}{2}} \quad \mathcal{N}^{4\sigma^2}_{-\frac{5\|\boldsymbol{\mu}\|}{2}} \quad \mathcal{N}^{\sigma^2}_{-\frac{\|\boldsymbol{\mu}\|}{2}} \quad \mathcal{N}^{\sigma^2}_{\frac{\|\boldsymbol{\mu}\|}{2}} \quad \mathcal{N}^{\sigma^2}_{-\frac{\|\boldsymbol{\mu}\|}{2}} \quad \mathcal{N}^{\sigma^2}_{-\frac{3\|\boldsymbol{\mu}\|}{2}}\right] \qquad \text{for } i,j \in C_1 \times C_0,$$

$$\left[\mathcal{N}^{4\sigma^2}_{-\frac{5\|\boldsymbol{\mu}\|}{2}} \quad \mathcal{N}^{4\sigma^2}_{-\frac{\|\boldsymbol{\mu}\|}{2}} \quad \mathcal{N}^{4\sigma^2}_{-\frac{5\|\boldsymbol{\mu}\|}{2}} \quad \mathcal{N}^{4\sigma^2}_{-\frac{\|\boldsymbol{\mu}\|}{2}} \quad \mathcal{N}^{\sigma^2}_{-\frac{\|\boldsymbol{\mu}\|}{2}} \quad \mathcal{N}^{\sigma^2}_{-\frac{3\|\boldsymbol{\mu}\|}{2}} \quad \mathcal{N}^{\sigma^2}_{-\frac{\|\boldsymbol{\mu}\|}{2}} \quad \mathcal{N}^{\sigma^2}_{\frac{\|\boldsymbol{\mu}\|}{2}}\right] \qquad \text{for } i,j \in C_{-1} \times C_0,$$

$$\left[\mathcal{N}^{4\sigma^2}_{-\frac{3\|\boldsymbol{\mu}\|}{2}} \quad \mathcal{N}^{4\sigma^2}_{-\frac{3\|\boldsymbol{\mu}\|}{2}} \quad \mathcal{N}^{4\sigma^2}_{-\frac{3\|\boldsymbol{\mu}\|}{2}} \quad \mathcal{N}^{4\sigma^2}_{-\frac{3\|\boldsymbol{\mu}\|}{2}} \quad \mathcal{N}^{\sigma^2}_{-\frac{\|\boldsymbol{\mu}\|}{2}} \quad \mathcal{N}^{\sigma^2}_{-\frac{\|\boldsymbol{\mu}\|}{2}} \quad \mathcal{N}^{\sigma^2}_{-\frac{\|\boldsymbol{\mu}\|}{2}} \quad \mathcal{N}^{\sigma^2}_{-\frac{\|\boldsymbol{\mu}\|}{2}}\right] \qquad \text{for } i,j \in C^2_0,$$

Next, we will use the following lemma regarding LeakyRelu concentration.

**Lemma 6** (Lemma A.6 in [10])**.** Fix $s \in \mathbb{N}$, and let $z_1, \ldots, z_s$ be jointly Gaussian random variables with marginals $\boldsymbol{z}_i \sim \mathcal{N}(\mu_i, \sigma_i^2)$. There exists an absolute constant $C > 0$ such that with probability at least $1 - o_s(1)$, we have

$$\text{LeakyRelu}(z_i) = \text{LeakyRelu}(\mu_i) \pm C\sigma_i\sqrt{\log s}, \quad \text{for all } i \in [s].$$

Using Lemma 6 with the assumption on $\|\boldsymbol{\mu}\|$ and a union bound, we have that with probability at least $1 - o_n(1)$, LeakyRelu($\boldsymbol{\Delta}_{ij}$) is (up to $1 \pm o(1)$)

$$\left[\frac{\|\boldsymbol{\mu}\|}{2} \quad \frac{-7\beta\|\boldsymbol{\mu}\|}{2} \quad -\frac{3\beta\|\boldsymbol{\mu}\|}{2} \quad -\frac{3\beta\|\boldsymbol{\mu}\|}{2} \quad \frac{\|\boldsymbol{\mu}\|}{2} \quad \frac{\|\boldsymbol{\mu}\|}{2} \quad -\frac{3\beta\|\boldsymbol{\mu}\|}{2} \quad -\frac{3\beta\|\boldsymbol{\mu}\|}{2}\right] \qquad \text{for } i,j \in C^2_1,$$

$$\left[-\frac{7\beta\|\boldsymbol{\mu}\|}{2} \quad \frac{\|\boldsymbol{\mu}\|}{2} \quad -\frac{3\beta\|\boldsymbol{\mu}\|}{2} \quad -\frac{3\beta\|\boldsymbol{\mu}\|}{2} \quad -\frac{3\beta\|\boldsymbol{\mu}\|}{2} \quad -\frac{3\beta\|\boldsymbol{\mu}\|}{2} \quad \frac{\|\boldsymbol{\mu}\|}{2} \quad \frac{\|\boldsymbol{\mu}\|}{2}\right] \qquad \text{for } i,j \in C^2_{-1},$$

$$\left[-\frac{3\beta\|\boldsymbol{\mu}\|}{2} \quad -\frac{3\beta\|\boldsymbol{\mu}\|}{2} \quad \frac{\|\boldsymbol{\mu}\|}{2} \quad -\frac{7\beta\|\boldsymbol{\mu}\|}{2} \quad -\frac{3\beta\|\boldsymbol{\mu}\|}{2} \quad \frac{\|\boldsymbol{\mu}\|}{2} \quad \frac{\|\boldsymbol{\mu}\|}{2} \quad -\frac{3\beta\|\boldsymbol{\mu}\|}{2}\right] \qquad \text{for } i,j \in C_1 \times C_{-1},$$

$$\left[-\frac{3\beta\|\boldsymbol{\mu}\|}{2} \quad -\frac{3\beta\|\boldsymbol{\mu}\|}{2} \quad -\frac{7\beta\|\boldsymbol{\mu}\|}{2} \quad \frac{\|\boldsymbol{\mu}\|}{2} \quad \frac{\|\boldsymbol{\mu}\|}{2} \quad -\frac{3\beta\|\boldsymbol{\mu}\|}{2} \quad -\frac{3\beta\|\boldsymbol{\mu}\|}{2} \quad \frac{\|\boldsymbol{\mu}\|}{2}\right] \qquad \text{for } i,j \in C_{-1} \times C_1,$$

$$\left[-\frac{\beta\|\boldsymbol{\mu}\|}{2} \quad -\frac{5\beta\|\boldsymbol{\mu}\|}{2} \quad -\frac{5\beta\|\boldsymbol{\mu}\|}{2} \quad -\frac{\beta\|\boldsymbol{\mu}\|}{2} \quad \frac{\|\boldsymbol{\mu}\|}{2} \quad -\frac{\|\boldsymbol{\mu}\|}{2} \quad -\frac{3\beta\|\boldsymbol{\mu}\|}{2} \quad -\frac{\|\beta\boldsymbol{\mu}\|}{2}\right] \qquad \text{for } i,j \in C_0 \times C_1,$$

$$\left[-\frac{5\beta\|\boldsymbol{\mu}\|}{2} \quad -\frac{\beta\|\boldsymbol{\mu}\|}{2} \quad -\frac{\beta\|\boldsymbol{\mu}\|}{2} \quad -\frac{5\beta\|\boldsymbol{\mu}\|}{2} \quad -\frac{3\beta\|\boldsymbol{\mu}\|}{2} \quad -\frac{\beta\|\boldsymbol{\mu}\|}{2} \quad \frac{\|\boldsymbol{\mu}\|}{2} \quad -\frac{\beta\|\boldsymbol{\mu}\|}{2}\right] \qquad \text{for } i,j \in C_0 \times C_{-1},$$

$$\left[-\frac{\beta\|\boldsymbol{\mu}\|}{2} \quad -\frac{5\beta\|\boldsymbol{\mu}\|}{2} \quad -\frac{\beta\|\boldsymbol{\mu}\|}{2} \quad -\frac{5\beta\|\boldsymbol{\mu}\|}{2} \quad -\frac{\beta\|\boldsymbol{\mu}\|}{2} \quad \frac{\|\boldsymbol{\mu}\|}{2} \quad -\frac{\beta\|\boldsymbol{\mu}\|}{2} \quad -\frac{3\beta\|\boldsymbol{\mu}\|}{2}\right] \qquad \text{for } i,j \in C_1 \times C_0,$$

$$\left[-\frac{5\beta\|\boldsymbol{\mu}\|}{2} \quad -\frac{\beta\|\boldsymbol{\mu}\|}{2} \quad -\frac{5\beta\|\boldsymbol{\mu}\|}{2} \quad -\frac{\beta\|\boldsymbol{\mu}\|}{2} \quad -\frac{\beta\|\boldsymbol{\mu}\|}{2} \quad -\frac{3\beta\|\boldsymbol{\mu}\|}{2} \quad -\frac{\|\beta\boldsymbol{\mu}\|}{2} \quad \frac{\|\boldsymbol{\mu}\|}{2}\right] \qquad \text{for } i,j \in C_{-1} \times C_0,$$

$$\left[-\frac{3\beta\|\boldsymbol{\mu}\|}{2} \quad -\frac{3\beta\|\boldsymbol{\mu}\|}{2} \quad -\frac{3\beta\|\boldsymbol{\mu}\|}{2} \quad -\frac{3\beta\|\boldsymbol{\mu}\|}{2} \quad -\frac{\beta\|\boldsymbol{\mu}\|}{2} \quad -\frac{\beta\|\boldsymbol{\mu}\|}{2} \quad -\frac{\beta\|\boldsymbol{\mu}\|}{2} \quad -\frac{\beta\|\boldsymbol{\mu}\|}{2}\right] \qquad \text{for } i,j \in C^2_0.$$

Then,

$$\boldsymbol{r}^T \cdot \text{LeakyRelu}(\boldsymbol{\Delta}_{ij}) = \begin{cases} 10R\beta\|\boldsymbol{\mu}\|(1 \pm o(1)) & \text{if } i,j \in C^2_1 \\ -2R\|\boldsymbol{\mu}\|(1 + 2\beta)(1 \pm o(1)) & \text{if } i,j \in C^2_{-1} \\ -2R\|\boldsymbol{\mu}\|(1 + 5\beta)(1 \pm o(1)) & \text{if } i \in C_1,\ j \in C_{-1} \\ 10R\beta\|\boldsymbol{\mu}\|(1 \pm o(1)) & \text{if } i \in C_{-1},\ j \in C_1 \\ -\frac{R}{2}\|\boldsymbol{\mu}\|(1 - 21\beta)(1 \pm o(1)) & \text{if } i \in C_0,\ j \in C_1 \\ -\frac{R}{2}\|\boldsymbol{\mu}\|(1 - 11\beta)(1 \pm o(1)) & \text{if } i \in C_0,\ j \in C_{-1} \\ -\frac{R}{2}\|\boldsymbol{\mu}\|(1 - 5\beta)(1 \pm o(1)) & \text{if } i \in C_1,\ j \in C_0 \\ -\frac{R}{2}\|\boldsymbol{\mu}\|(1 - 5\beta)(1 \pm o(1)) & \text{if } i \in C_{-1},\ j \in C_0 \\ 2R\beta\|\boldsymbol{\mu}\|(1 \pm o(1)) & \text{if } i,j \in C^2_0 \end{cases},$$

and the proof is complete. $\qquad\square$

Next we will define our high probability event.

**Definition 2.** $\mathcal{E}' \overset{\text{def}}{=} \mathcal{E} \cap \mathcal{E}^*$, where $\mathcal{E}^*$ is the event that for a fixed $\boldsymbol{w} \in \mathbb{R}^d$, all $i \in [n]$ satisfy $|\boldsymbol{w}^T \mathbf{X}_i - \mathbf{E}[\boldsymbol{w}^T \mathbf{X}_i]| \leq 10\sigma \|\boldsymbol{w}\|_2 \sqrt{\log n}$.

The following lemma is obtained by using Lemma 4 with standard Gaussian concentration and a union bound.

**Lemma 7.** With probability at least $1 - 1/\mathrm{poly}(n)$ event $\mathcal{E}'$ holds.

**Corollary 8.** Suppose that $p, q$ satisfy Assumption 1, $\|\boldsymbol{\mu}\| = \omega(\sigma\sqrt{\log n})$ and fix $R \in \mathbb{R}$. Then, there exists a choice of attention architecture $\Psi$ such that with probability $1 - o_n(1)$ over $(\mathbf{A}, \mathbf{X}) \sim$ $\mathsf{CSBM}(n, p, q, \boldsymbol{\mu}, \sigma^2)$ it holds that

$$
\gamma_{ij} = \begin{cases}
\frac{3}{np}(1 \pm o(1)) & \text{if } i,j \in C_0^2 \cup C_1^2 \\
\frac{3}{nq}(1 \pm o(1)) & \text{if } i,j \in C_{-1} \times C_1 \\
\frac{3}{nq}\exp(-\Theta(R\|\boldsymbol{\mu}\|)) & \text{if } i,j \in C_{-1} \times C_{-1} \cup C_0 \\
\frac{3}{np}\exp(-\Theta(R\|\boldsymbol{\mu}\|)) & \text{otherwise}
\end{cases},
$$

where $R$ is a parameter of the architecture.

**Proof.** The proof is immediate. First applying the ansatz from Lemma 5 with $\beta < 1/25$, Lemma 7 and a union bound. Using the definition of $\gamma_{ij}$ concludes the proof. $\qquad\square$

Next, we prove Thm. 1 that the model distinguish nodes from $C_0$ for any choice of $p, q$ satisfying Assumption 1. We restate the theorem for convince.

**Theorem 9** (Formal restatement of Thm. 1). Suppose that $p, q$ satisfy Assumption 1 and $\|\boldsymbol{\mu}\|_2 = \omega(\sigma\sqrt{\log n})$. Then, there exists a choice of attention architecture $\Psi$ such that with probability at least $1 - o_n(1)$ over the data $(\mathbf{X}, \mathbf{A}) \sim \mathsf{CSBM}(n, p, q, \boldsymbol{\mu}, \sigma^2)$, the estimator

$$
\hat{x}_i \overset{\text{def}}{=} \sum_{j \in N_i} \gamma_{ij} \tilde{\boldsymbol{w}}^T \mathbf{X}_j + b \quad \text{where } \tilde{\boldsymbol{w}} = \boldsymbol{\mu}/\|\boldsymbol{\mu}\|, \ b = -\|\boldsymbol{\mu}\|/2
$$

satisfies $\hat{x}_i < 0$ if and only if $i \in C_0$.

**Proof.** Let $\Psi$ be the architecture from Cor. 8 and let $R$ satisfy $R\|\boldsymbol{\mu}\|_2 = \omega(1)$. We will compute the mean and variance of the estimator $\hat{x}_i$ conditioned on $\mathcal{E}'$. Suppose that $i \in C_0$. By using Cor. 8, Definition 2 and our assumption on $\|\boldsymbol{\mu}\|$ and $R$, we have

$$
\max\left\{ \frac{3}{np}\exp(-\Theta(R\|\boldsymbol{\mu}\|)), \frac{3}{nq}\exp(-\Theta(R\|\boldsymbol{\mu}\|)) \right\} = o\left( \frac{1}{n(p+2q)} \right),
$$

and therefore

$$
\begin{aligned}
\mathbf{E}\left[ \hat{x}_i \mid \mathcal{E}' \right] &= \mathbf{E}\left[ \sum_{k \in \{-1,0,1\}} \sum_{j \in N_i \cap C_k} \gamma_{ij} \tilde{\boldsymbol{w}}^T \mathbf{X}_j \mid \mathcal{E}' \right] - \frac{\|\boldsymbol{\mu}\|}{2} \\
&= \mathbf{E}[|C_0 \cap N_i| \mid \mathcal{E}']\left( \pm\frac{3}{np}(1 \pm o(1)) \cdot 10\sigma\sqrt{\log n} \right) \\
&\quad + \mathbf{E}[|C_1 \cap N_i| \mid \mathcal{E}']\left( o\left( \frac{1}{n(p+2q)} \right) \cdot (\|\boldsymbol{\mu}\| \pm 10\sigma\sqrt{\log n}) \right) \\
&\quad + \mathbf{E}[|C_{-1} \cap N_i| \mid \mathcal{E}']\left( o\left( \frac{1}{n(p+2q)} \right) \cdot (-\|\boldsymbol{\mu}\| \pm 10\sigma\sqrt{\log n}) \right) - \frac{\|\boldsymbol{\mu}\|}{2} \\
&= -\frac{\|\boldsymbol{\mu}\|}{2}(1 \pm o(1)).
\end{aligned}
$$

By similar reasoning we have that for $i \in C_{-1} \cup C_1$, $\mathbf{E}\left[\hat{x}_i \mid \boldsymbol{\mathcal{E}}'\right] = \frac{\|\boldsymbol{\mu}\|}{2}(1 \pm o(1))$.

Next, we claim that for each $i \in [n]$ the random variable $\hat{x}_i$ given $\boldsymbol{\mathcal{E}}'$ is sub-Gaussian with a small sub-Gaussian constant compared to the above expectation. The following lemma is a straightforward adaptation of Lemma A.11 in [10], and we provide its proof for completeness.

**Lemma 10.** Conditioned on $\boldsymbol{\mathcal{E}}'$, the random variables $\{\hat{x}_i\}_i$ are sub-Gaussian with parameter $\tilde{\sigma}_i^2 = O\left(\frac{\sigma^2}{np}\right)$ if $i \in C_0 \cup C_1$ and $\tilde{\sigma}_i^2 = O\left(\frac{\sigma^2}{nq}\right)$ otherwise.

**Proof.** Fix $i \in [n]$, and write $\mathbf{X}_i = \varepsilon_i \boldsymbol{\mu} + \sigma \boldsymbol{g}_i$ where $\boldsymbol{g}_i \sim \mathcal{N}(0, \mathbf{I}_d)$, and $\varepsilon_i$ denotes the class membership. Consider $\hat{x}_i$ as a function of $\boldsymbol{g} = [\boldsymbol{g}_1 \circ \boldsymbol{g}_2 \circ \cdots \circ \boldsymbol{g}_n] \in \mathbb{R}^{nd}$, where $\circ$ denotes vertical concatenation. Namely, consider the function

$$\hat{x}_i = f_i(\boldsymbol{g}) \overset{\text{def}}{=} \sum_{j \in N_i} \gamma_{ij}(\boldsymbol{g})\, \tilde{\boldsymbol{w}}^T (\varepsilon_j \boldsymbol{\mu} + \sigma \boldsymbol{g}_j) - \|\boldsymbol{\mu}\|/2, \quad i \in [n].$$

Since $\boldsymbol{g} \sim \mathcal{N}(0, \mathbf{I}_{nd})$, proving that $\hat{x}_i$ given $\boldsymbol{\mathcal{E}}'$ is sub-Gaussian for each $i \in [n]$, reduces to showing that the function $f_i : \mathbb{R}^{nd} \to \mathbb{R}$ is Lipschitz over $E \subseteq \mathbb{R}^{nd}$ defined by $\boldsymbol{\mathcal{E}}'$ and the relation $\mathbf{X}_i = \varepsilon_i \boldsymbol{\mu} + \sigma \boldsymbol{g}_i$. That is, $E \overset{\text{def}}{=} \left\{\boldsymbol{g} \in \mathbb{R}^{nd} \mid |\tilde{\boldsymbol{w}}^T \boldsymbol{g}_i| \leq 10\sqrt{\log n}, \forall i \in [n]\right\}$. Specifically, we show that conditioning on the event $\boldsymbol{\mathcal{E}}'$ (which restricts $\boldsymbol{g} \in E$), the Lipschitz constant $L_{f_i}$ of $f_i$ satisfies $L_{f_i} = O\left(\frac{\sigma}{\sqrt{np}}\right)$ for $i \in C_0 \cup C_1$ and $L_{f_i} = O\left(\frac{\sigma}{nq}\right)$ otherwise, and hence proving the claim.

To compute the Lipschitz constant of $f_i(\boldsymbol{g})$ for $i \in [n]$, let us denote $\mathbf{X} = [\mathbf{X}_1 \circ \mathbf{X}_2 \circ \cdots \circ \mathbf{X}_n]$ and consider the function

$$\tilde{f}_i(\mathbf{X}) \overset{\text{def}}{=} \sum_{j \in N_i} \gamma_{ij}(\mathbf{X})\, \tilde{\boldsymbol{w}}^T \mathbf{X}_j, \quad i \in [n]$$

Let us assume without loss of generality that $i \in C_0$ (the cases for $i \in C_1$ and $i \in C_{-1}$ are obtained identically). Conditioning on the event $\boldsymbol{\mathcal{E}}'$, which imposes the restriction that $\mathbf{X} \in \tilde{E}$ where

$$\tilde{E} \overset{\text{def}}{=} \left\{\mathbf{X} \in \mathbb{R}^{nd} \mid |\mathbf{X}_i - \varepsilon_i \boldsymbol{\mu}| \leq 10\sigma\sqrt{\log n}, \forall i \in [n]\right\}.$$

Conditioning on $\boldsymbol{\mathcal{E}}'$ (which restricts $\mathbf{X}, \mathbf{X}' \in \tilde{E}$), using Cor. 8 and recalling that $R$ satisfies $R\|\boldsymbol{\mu}\|_2 = \omega(1)$, we get[3]

$$\left|\tilde{f}_i(\mathbf{X}) - \tilde{f}_i(\mathbf{X}')\right|$$

$$\simeq \left| \sum_{j \in N_i \cap C_0} \frac{3}{np} \tilde{\boldsymbol{w}}^T (\mathbf{X}_j - \mathbf{X}'_j) + \sum_{j \in N_i \cap C_1} \frac{3}{np} \cdot e^{-\Theta(R\|\boldsymbol{\mu}\|_2)} \tilde{\boldsymbol{w}}^T (\mathbf{X}_j - \mathbf{X}'_j) + \sum_{j \in N_i \cap C_{-1}} \frac{3}{np} \cdot e^{-\Theta(R\|\boldsymbol{\mu}\|_2)} \tilde{\boldsymbol{w}}^T (\mathbf{X}_j - \mathbf{X}'_j) \right|$$

$$= \left| \begin{bmatrix} \frac{3}{np}(1 \pm o(1))\tilde{\boldsymbol{w}} & \text{if } j \in N_i \cap C_0 \\ \frac{3}{np}\exp(-\Theta(R\|\boldsymbol{\mu}\|_2))(1 \pm o(1))\tilde{\boldsymbol{w}} & \text{if } j \in N_i \cap C_1 \\ \frac{3}{np}\exp(-\Theta(R\|\boldsymbol{\mu}\|_2))(1 \pm o(1))\tilde{\boldsymbol{w}} & \text{if } j \in N_i \cap C_{-1} \\ 0 & \text{if } j \notin N_i \end{bmatrix}_{j \in [n]}^T (\mathbf{X} - \mathbf{X}') \right|$$

$$\leq \left\| \begin{bmatrix} \frac{3}{np}(1 \pm o(1))\tilde{\boldsymbol{w}} & \text{if } j \in N_i \cap C_0 \\ \frac{3}{np}\exp(-\Theta(R\|\boldsymbol{\mu}\|_2))(1 \pm o(1))\tilde{\boldsymbol{w}} & \text{if } j \in N_i \cap C_1 \\ \frac{3}{np}\exp(-\Theta(R\|\boldsymbol{\mu}\|_2))(1 \pm o(1))\tilde{\boldsymbol{w}} & \text{if } j \in N_i \cap C_{-1} \\ 0 & \text{if } j \notin N_i \end{bmatrix}_{j \in [n]} \right\|_2 \|\mathbf{X} - \mathbf{X}'\|_2$$

$$\leq \sqrt{\frac{3}{np}}(1 + o(1))\|\tilde{\boldsymbol{w}}\|_2 \|\mathbf{X} - \mathbf{X}'\|_2$$

$$= \sqrt{\frac{3}{np}}(1 + o(1)) \|\mathbf{X} - \mathbf{X}'\|_2.$$

---

[3]We drop the $(1 \pm o(1))$ in the first line of the computation for compactness and use $\simeq$ as notation.

This shows the Lipschitz constant of $\tilde{f}_i(\mathbf{X})$ over $\tilde{E}$ satisfies $L_{\tilde{f}_i} = O\left(\frac{1}{\sqrt{np}}\right)$. On the other hand, by viewing $\mathbf{X}$ as a function of $\boldsymbol{g}$, it is straightforward to see that the function $h(\boldsymbol{g}) : \mathbb{R}^{nd} \to \mathbb{R}^{nd}$ defined by $h(\boldsymbol{g}) \overset{\text{def}}{=} \mathbf{X}(\boldsymbol{g})$ has Lipschitz constant $L_h = \sigma$, as

$$\|h(\boldsymbol{g}) - h(\boldsymbol{g}')\|_2 = \|\boldsymbol{\varepsilon}\boldsymbol{\mu} + \sigma\boldsymbol{g} - (\boldsymbol{\varepsilon}\boldsymbol{\mu} + \sigma\boldsymbol{g}')\|_2 = \sigma\|\boldsymbol{g} - \boldsymbol{g}'\|_2.$$

Therefore, since $f_i(\boldsymbol{g}) = \tilde{f}_i(h(\boldsymbol{g}))$ and $\boldsymbol{g} \in E$ if and only if $\mathbf{X} \in \tilde{E}$, we have that, conditioning on $\mathcal{E}'$, the function $\hat{x}_i = f_i(\boldsymbol{g})$ is Lipschitz continuous with Lipschitz constant $L_{f_i} = L_{\tilde{f}_i} L_h = O\left(\frac{\sigma}{\sqrt{np}}\right)$. Since $\boldsymbol{g} \sim \mathcal{N}(0, \mathbf{I}_{nd})$, we know that $\hat{x}_i$ is sub-Gaussian with sub-Gaussian constant $\tilde{\sigma}^2 = L_{f_i}^2 = O\left(\frac{\sigma^2}{np}\right)$. $\qquad\square$

The following lemma will be used for bounding the misclassification probability.

**Lemma 11** ([19])**.** Let $x_1, \ldots, x_n$ be sub-Gaussian random variables with the same mean and sub-Gaussian parameter $\tilde{\sigma}^2$. Then,

$$\mathbf{E}\left[\max_{i \in [n]} (x_i - \mathbf{E}[x_i])\right] \le \tilde{\sigma}\sqrt{2 \log n}.$$

Moreover, for any $t > 0$

$$\mathbf{Pr}\left[\max_{i \in [n]} (x_i - \mathbf{E}[x_i]) > t\right] \le 2n \exp\left(-\frac{t^2}{2\tilde{\sigma}^2}\right).$$

We bound the probability of misclassification

$$\mathbf{Pr}\left[\max_{i \in C_0} \hat{x}_i \ge 0\right] \le \mathbf{Pr}\left[\max_{i \in C_0} \hat{x}_i > t + \mathbf{E}[\hat{x}_i]\right],$$

for $t < |\mathbf{E}[\hat{x}_i]| = \frac{\|\boldsymbol{\mu}\|_2}{2}(1 \pm o(1))$. By Lemma 10, picking $t = \Theta\left(\sigma\sqrt{\log |C_0|}\right)$ and applying Lemma 11 implies that the above probability is $1/\text{poly}(n)$.

Similarly for class $C_1 \cup C_{-1}$ we have that the misclassification probability is

$$\mathbf{Pr}\left[\min_{i \in C_1 \cup C_{-1}} \hat{x}_i \le 0\right] = \mathbf{Pr}\left[-\max_{i \in C_1 \cup C_{-1}} (-\hat{x}_i) \le 0\right] = \mathbf{Pr}\left[\max_{i \in C_1 \cup C_{-1}} (-\hat{x}_i) \ge 0\right]$$
$$\le \mathbf{Pr}\left[\max_{i \in C_1 \cup C_{-1}} -\hat{x}_i > t - \mathbf{E}[\hat{x}_i]\right],$$

for $t < \mathbf{E}[\hat{x}_i]$. Picking $t = \Theta\left(\sigma\sqrt{\log |C_1 \cup C_{-1}|}\right)$ and applying Lemma 11 and a union bound over the misclassification probabilities of both classes conclude the proof of the corollary. $\qquad\square$

Combining Thm. 9 with Lemma 3, we immediately get Cor. 2 which we restate below.

**Corollary 12.** Suppose $p, q = \Omega(\log^2 n/n)$ and $\|\boldsymbol{\mu}\| \ge \omega\left(\sigma\sqrt{\frac{(p+2q)\log n}{n(p-q)^2}}\right)$. Then, there is a choice of attention architecture $\Psi$ such that, with probability at least $1 - o(1)$ over the data $(\mathbf{X}, \mathbf{A}) \sim$ CSBM$(n, p, q, \boldsymbol{\mu}, \sigma^2)$, CAT separates nodes $C_0$ from $C_1 \cup C_{-1}$.

# B Synthetic experiments

In this section, we present the complete results for the synthetic data experiments of §6. First, we describe the parameterization we use for the 1-layer GCN, GAT, and CAT models; then, we verify the behavior of the normalized score function ($\gamma_{ij}$) matches that of the theory presented in Cor. 8. In particular, we visualize the average of the following three groups of gammas (Fig. 2):

- Gammas $\gamma_{ij}$ included in $i, j \in C_0^2 \cup C_1^2$. Solid lines.
- Gammas $\gamma_{ij}$ included in $i, j \in C_{-1} \times C_1$. Dashed lines.
- The rest of gammas. Dotted lines.

For completeness, we also include the empirical results that validate Thm. 1 and Cor. 2, which were discussed already in §6.

**Experimental setup.** We assume the following parametrization for the 1-layer GCN, GAT, and CAT:

$$h_i' = \left( \sum_{j \in N_i} \gamma_{ij} \tilde{w}^T \mathbf{X}_j \right) - C \cdot \|\mu\|/2 \,, \tag{6}$$

where $N_i$ are the set of neighbors of node $i$, $\mathbf{X}_j$ are the features of node $j$—obtained from the CSBM described in §4, and $h_i'$ are the logits of the prediction of node $i$. Note that for GCN we have $\gamma_{ij} = \frac{1}{|N_i^*|}$. Otherwise, we consider the following parameterization of the score function $\Psi$:

$$\gamma_{ij} = \frac{\exp\left(\Psi(h_i, h_j)\right)}{\sum_{k \in N_i^*} \exp(\Psi(h_i, h_k))} \quad \text{where} \tag{7}$$

$$\Psi(h_i, h_j) \stackrel{\text{def}}{=} r^T \cdot \text{LeakyRelu}\left( \mathbf{S} \cdot \begin{bmatrix} \tilde{w}^T h_i \\ \tilde{w}^T h_j \end{bmatrix} + b \right) \,. \tag{8}$$

For these experiments, we define the parameters $\tilde{w}$, $\mathbf{S}$, $b$ and $r$ as in the proofs in App. A:

$$\tilde{w} \stackrel{\text{def}}{=} \frac{\mu}{\|\mu\|}, \qquad \mathbf{S} \stackrel{\text{def}}{=} \begin{bmatrix} 1 & 1 \\ -1 & -1 \\ 1 & -1 \\ -1 & 1 \\ 0 & 1 \\ 1 & 0 \\ 0 & -1 \\ -1 & 0 \end{bmatrix}, \qquad b \stackrel{\text{def}}{=} \begin{bmatrix} -3/2 \\ -3/2 \\ -3/2 \\ -3/2 \\ -1/2 \\ -1/2 \\ -1/2 \\ -1/2 \end{bmatrix} \cdot \|\mu\| \cdot C, \qquad r \stackrel{\text{def}}{=} R \cdot \begin{bmatrix} 2 \\ -2 \\ -2 \\ 2 \\ -1 \\ -1 \\ -1 \\ -1 \end{bmatrix}, \tag{9}$$

where $R > 0$ and $C > 0$ are arbitrary scaling parameters. Both $C$ and $R$ and input to the score function are set different for each of the models, as indicated in Table 2. In particular, we set $R = \frac{7}{\|\mu\|}$ for both GAT and CAT such that: i) all $\gamma_{ij}$ are distinguishable as we decrease $\|\mu\|$; and ii) we avoid numerical instabilities in the implementation when computing the exponential of $R \times \|\mu\|$ in order to obtain $\gamma_{ij}$ (see Cor. 8), as the exponential of small or large values leads to under/overflow issues. As for $C$, we set $C = 1$ for GAT and $C = (p - q)/(p + 2q)$ for CAT such that we counteract the fact that the distance between classes shrink as we increase $q$, see Lemma 3:

Table 2: Parameters for the synthetic experiments.

| Model | $C$ | $R$ | $h_i$ |
|-------|-----|-----|-------|
| GCN | 0 | — | — |
| GAT | 1 | $\frac{7}{\|\mu\|}$ | $\mathbf{X}_i$ |
| CAT | $\frac{p-q}{p+2q}$ | $\frac{7}{\|\mu\|}$ | $\frac{1}{|N_i^*|} \sum_{k \in N_i^*} \mathbf{X}_k$ |

Regarding the data model, we set $n = 10000$, $p = 0.5$, $\sigma = 0.1$, and $d = n / \left(5 \log^2(n)\right)$. We set the slope of the LeakyReLU activation to $\beta = 1/5$ for the GAT and $\beta = 0.01$ for CAT, such that the

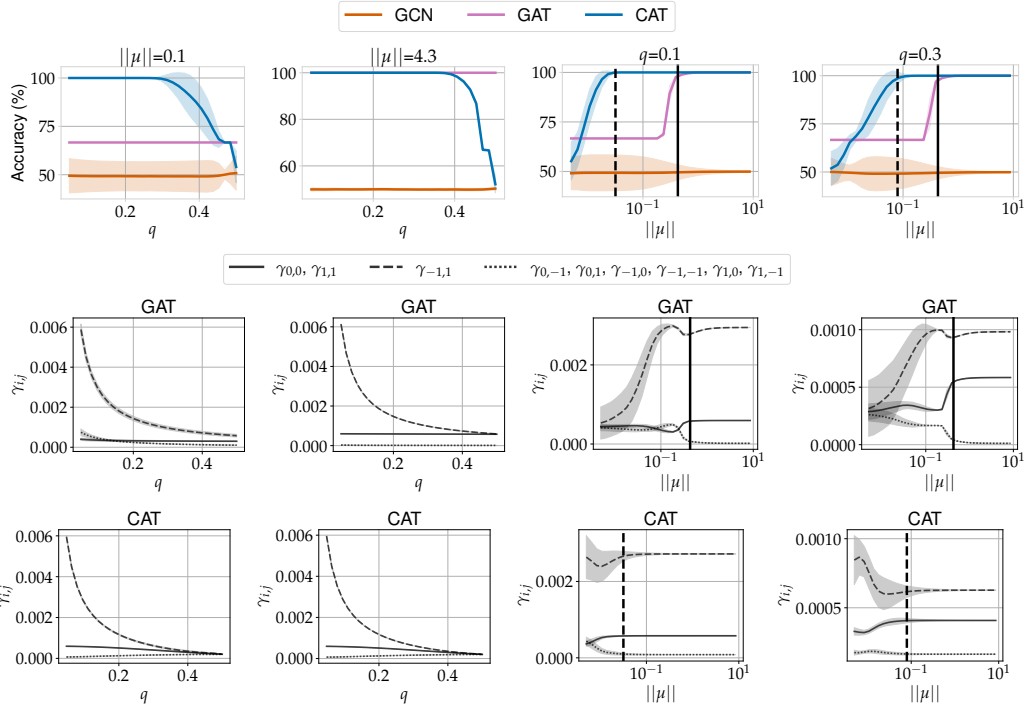

Figure 2: Synthetic data results. On the top row, we show the node classification, and in the following two rows we show the $\gamma_{ij}$ values for GAT and CAT respectively. In the two left-most figures, we show how the results vary with the noise level $q$ for $\|\boldsymbol{\mu}\| = 0.1$ and $\|\boldsymbol{\mu}\| = 4.3$. In the two right-most figures, we show how the results vary with the norm of the means $\|\boldsymbol{\mu}\|$ for $q = 0.1$ and $q = 0.3$. We use two vertical lines to present the classification threshold stated in Thm. 1 (solid line) and Cor. 2 (dashed line).

proof of Cor. 8 is valid. To assess the sensitivity to structural noise, we present the complete results for two sets of experiments. First, we vary the noise level $q$ between 0 and 0.5, fixing the mean vector $\boldsymbol{\mu}$. We test two values of $\|\boldsymbol{\mu}\|$: the first corresponds to the *easy* regime ($\|\boldsymbol{\mu}\| = 10\sigma\sqrt{2\log n} \approx 4.3$) where classes are far apart; and the second correspond to the *hard* regime ($\|\boldsymbol{\mu}\| = \sigma = 0.1$) where the distance between the clusters is small. In the second experiment we modify instead the distance between the means, sweeping $\|\boldsymbol{\mu}\|$ in the range $\left[\sigma/20, 20\sigma\sqrt{2\log n}\right]$ which corresponds to $[0.005, 8.58]$, and thus covering the transition from the hard setting (small $\|\boldsymbol{\mu}\|$) to the easy one (large $\|\boldsymbol{\mu}\|$). In these experiments, we fix $q$ to 0.1 (low noise) and 0.3 (high noise).

**Results** are summarized in Fig. 2. The top row contains the node classification performance for each of the models (i.e., Fig. 1), the next two rows contain the $\gamma_{ij}$ values for GAT and CAT respectively. The two left-most columns of Fig. 2 show the results for the hard and easy regimes, respectively, as we vary the noise level $q$. In the hard regime, we observe that GAT is unable to achieve separation for any value of $q$, whereas CAT achieves perfect classification when $q$ is small enough. The gamma plots help shed some light on this question. For GAT, we observe that the gammas represented with the dotted and solid lines collapse for any value of $q$ (see middle plot), while this does not happen for CAT when the noise level is low (see bottom plot). This exemplifies the advantage of CAT over GAT as stated in Cor. 2. When the distance between the means is large enough, we see that GAT achieves perfect results independently of $q$, as stated in Thm. 1. We also observe that, in this case, the gammas represented with the dotted and solid lines do not collapse for any value of $q$. In contrast, as we increase $q$, CAT fails to satisfy the condition in Cor. 2, and therefore achieves inferior performance. We note that the low performance is due to the fact that all gammas collapse to the same value for large noise levels.

For the second set of experiments (two right-most columns of Fig. 1), where we fix $q$ and sweep $\|\boldsymbol{\mu}\|$, we observe that, for both values of $q$, there exists a transition in the accuracy of both GAT and CAT as a function of $\|\boldsymbol{\mu}\|$. As shown in the main manuscript, GAT achieves perfect accuracy when the

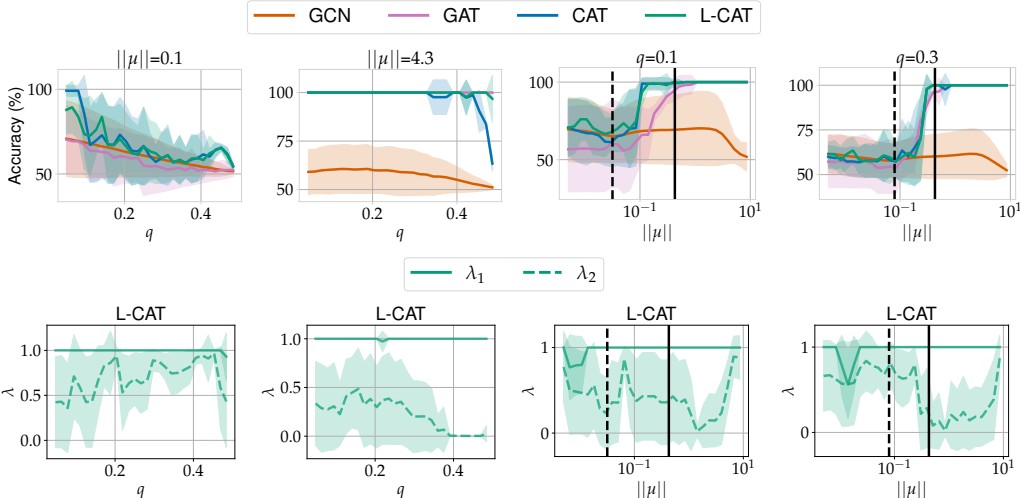

Figure 3: Synthetic data results learning $C$, $\lambda_1$ and $\lambda_2$. On the top row, we show the node classification accuracy, and in the bottom row we show the learned values of $\lambda_1$ and $\lambda_2$ for L-CAT. In the two left-most figures, we show how the results vary with the noise level $q$ for $\|\boldsymbol{\mu}\| = 0.1$ and $\|\boldsymbol{\mu}\| = 4.3$. In the two right-most figures, we show how the results vary with the norm of the means $\|\boldsymbol{\mu}\|$ for $q = 0.1$ and $q = 0.3$. We use two vertical lines to present the classification threshold stated in Thm. 1 (solid line) and Cor. 2 (dashed line).

distance between the means satisfies the condition in Thm. 1 (solid vertical line in Fig. 1). Moreover, we can see the improvement CAT obtains over GAT. Indeed, when $\|\boldsymbol{\mu}\|$ satisfies the conditions of Cor. 2 (dashed vertical line in Fig. 1), the classification threshold is improved. As we increase $q$, we see that the gap between the two vertical lines decreases, which means that the improvement decreases as $q$ increments, exactly as stated in Cor. 2. This transition from the hard regime to the easy regime is also observed in the gamma plots: we observe the largest difference in value between the different groups of lambdas for values of $\|\boldsymbol{\mu}\|$ that satisfy the condition in Thm. 1 (that is to the right of the vertical lines).

## B.1 Other experiments

In the following, we extend the results for the synthetic data presented above. In particular, we aim to evaluate if L-CAT is able to achieve top performance regardless of the scenario. That is, we want to evaluate if L-CAT consistently performs at least as good as the best-performing model. We change the fixed-parameter setting of the previous section and, instead, we evaluate the performance of GCN, GAT, CAT and L-CAT when we learn the model-dependent parameters.

**Experimental setup.** We assume the same parametrization for the 1-layer GCN, GAT and CAT described in Eq. 6 and Eq. 8. For L-CAT, we add the parameters $\lambda_1$ and $\lambda_2$, as indicated in Eq. 5. We fix the parameters shared among the models, that is, $\tilde{\boldsymbol{w}}$, $\mathbf{S}$, $\boldsymbol{b}$, $\boldsymbol{r}$, and $R$, with the values indicated in Eq. 9. Different from previous experiments, we now learn $C$ and, for L-CAT, we also learn $\lambda_1$ and $\lambda_2$. We choose to fix part of the parameters (instead of learning them all) to keep the problem as similar as possible to the theoretical analysis we provided in §4 and App. A. If we instead learn all the parameters, it takes a single dimension of the features to be (close to) linearly separable to find a solution that achieves a similar performance regardless of the model, which hinders the analysis. This is a consequence of the probabilistic nature of the features. One way of solving this issue would be to make $n$ big enough. Instead, we opt to have a fixed $n$ and reduce the degrees of freedom of the models by fixing the parameters shared across all models. The rest of the experimental setup matches the one from App. B. Additionally, we use the Adam optimizer [16] with a learning rate of 0.05, and we train for 100.

**Results** are summarized in Fig. 3. The top row contains the node classification performance for every model, while the bottom row contains the learned values of $\lambda_1$ (solid line) and $\lambda_2$ (dashed line) with L-CAT. The two left-most columns of Fig. 3 show the results for the hard and easy regimes,

respectively, as we vary the noise level $q$. In the hard regime, we see rather noisy results. Still, the behaviour is similar to that of Fig. 2: the performance of CAT degrades as we increase $q$. We also observe that, on average, CAT outperforms GAT. In this case, we observe that L-CAT achieves similar performance as CAT, which can be explained by inspecting the learned values of lambda in the bottom row. We observe that $\lambda_1 = 1$ and $\lambda_2 \geq 0.5$ on average for all values of $q$. This indicates that L-CAT is closer to CAT than to GAT. When the distance between the means is large enough (i.e., $\|\boldsymbol{\mu}\| = 4.3$), we see that GAT achieves perfect results independently of $q$ while the performance of CAT deteriorates with large values of $q$, the same trend as in Fig. 2. Remarkably, we observe that L-CAT also achieves perfect results independently of $q$. If we inspect the lambda values, we first see that $\lambda = 1$ for all $q$, thus the interpolation happens between CAT and GAT. Looking at the values of $\lambda_2$, we observe that, for small values of $q$, $\lambda_2$ is pretty noisy, which is expected since any solution achieves perfect performance. Interestingly, we have that $\lambda_2 = 0$ for large values of $q$, with negligible variance. This indicates that L-CAT learns that it must behave like GAT in order to perform well.

For the second set of experiments (two right-most columns of Fig. 3), we fix $q$ and sweep $\|\boldsymbol{\mu}\|$ like we did in Fig. 2. Here, we observe a similar trend: for both values of $q$, there exists a transition in the accuracy of both GAT and CAT as a function of $\|\boldsymbol{\mu}\|$. Yet once again, we observe that L-CAT consistently achieves a similar performance to the best-performing model in every situation.

## C   Dataset description

We present further details about the datasets used in our experiments, summarized in Table 3. All datasets are undirected (or transformed to undirected otherwise) and transductive.

The upper rows of the table refer to datasets used in §6 taken from the PyTorch Geometric framework.[4] The following paragraphs present a short description of such datasets.

**Amazon Computers & Photos** are datasets taken from [23], in which nodes represent products, and edges indicate that the products are usually bought together. The node features are a Bag of Words (BoW) representation of the product reviews. The task is to predict the category of the products.

**GitHub** is a dataset introduced in [20], in which nodes correspond to developers, and edges indicate mutual follow relationship. Node features are embeddings extracted from the developer's starred repositories and profile information (e.g., location or employer). The task is to infer whether a node relates to web or machine learning development.

**FacebookPagePage** is a dataset introduced in [20], where nodes are Facebook pages, and edges imply mutual likes between the pages. Nodes features are text embeddings extracted from the pages' description. The task consist on identifying the page's category.

**TwitchEN** is a dataset introduced in [20]. Here, nodes correspond to Twitch gamers, and links reveal mutual friendship. Node features are an embedding of games liked, location, and streaming habits. The task is to infer if a gamer uses explicit content.

**Coauthor Physics & CS** are datasets introduced in [23]. In this case, nodes represent authors which are connected with an edge if they have co-authored a paper. Node features are BoW representations of the keywords of the author's papers. The task consist on mapping each author to their corresponding field of study.

**DBLP** is a dataset introduced in [6] that represents a citation network. In this dataset, nodes represent papers and edges correspond to citations. Node features are BoW representations of the keywords of the papers. The task is to predict the research area of the papers.

**PubMed, Cora & CiteSeer** are citation networks introduced in [31]. Nodes represent documents, and edges refer to citations between the documents. Node features are BoW representations of the documents. The task is to infer the topic of the documents.

The bottom rows of Table 3 refer to the datasets from Open Graph Benchmark (OGB) [13].[5] We include a short description of them in the paragraphs below.

---

[4] https://pytorch-geometric.readthedocs.io/en/latest/modules/datasets.html
[5] https://ogb.stanford.edu/docs/nodeprop

**ogbn-arxiv** is a citation network of computer science papers in arXiv [27]. Nodes represent papers, and directed edges refer to citations among them. Node features are embeddings of the title and abstract of the papers. The task is to predict the research area of the nodes.

**ogbn-products** contains a co-purchasing network [5]. Nodes represent products, and links are present whenever two products are bought together. Node features are embeddings of a BoW representation of the product description. The task is to infer the category of the products.

**ogbn-mag** is a heterogeneous network formed from a subgraph of the Microsoft Academic Graph (MAG) [27]. Nodes can belong to one of these four types: authors, papers, institutions and fields of study. Moreover, directed edges belong to one of the following categories: "author is affiliated with an institution," "author has written a paper," "paper cites a paper," and "paper belongs to a research area." Only nodes that are papers contain node features, which are a text embedding of the document content. The task is to predict the venue of the nodes that are papers.

**ogbn-proteins** is a network whose nodes represent proteins and edges indicate different types of associations among them. This dataset does not contain node features. The tasks are to predict multiple protein functions, each of them being a binary classification problem.

Table 3: Dataset statistics. On the top part of the table, we show the datasets used in §6. On the bottom part of the table, we show the datasets from the OGB suite.

| Name | #Nodes | #Edges | Avg. degree | #Node feats. | #Edge feats. | #Tasks | Task Type |
|------|--------|--------|-------------|--------------|--------------|--------|-----------|
| AmazonComp. | 13,752 | 491,722 | 35.76 | 767 | - | 1 | 10-class clf. |
| AmazonPhoto | 7,650 | 238,162 | 31.13 | 745 | - | 1 | 8-class clf. |
| GitHub | 37,700 | 578,006 | 15.33 | 128 | - | 1 | Binary clf. |
| FacebookP. | 22,470 | 342,004 | 15.22 | 128 | - | 1 | 4-class clf. |
| CoauthorPh. | 34,493 | 495,924 | 14.38 | 8415 | - | 1 | 5-class clf. |
| TwitchEN | 7,126 | 77,774 | 10.91 | 128 | - | 1 | Binary clf. |
| CoauthorCS | 18,333 | 163,788 | 8.93 | 6805 | - | 1 | 15-class clf. |
| DBLP | 17,716 | 105,734 | 5.97 | 1639 | - | 1 | 4-class clf. |
| PubMed | 19,717 | 88,648 | 4.50 | 500 | - | 1 | 3-class clf. |
| Cora | 2,708 | 10,556 | 3.90 | 1433 | - | 1 | 7-class clf. |
| CiteSeer | 3,327 | 9,104 | 2.74 | 3703 | - | 1 | 6-class clf. |
| ogbn-arxiv | 169,343 | 1,166,243 | 6.89 | 128 | - | 1 | 40-class clf. |
| ogbn-products | 2,449,029 | 123,718,280 | 50.52 | 100 | - | 1 | 47-class clf. |
| ogbn-mag | 1,939,743 | 21,111,007 | 18.61 | 128 | 4 | 1 | 349-class clf. |
| ogbn-proteins | 132,534 | 79,122,504 | 597.00 | - | 8 | 112 | Multi-task |

# D   Real data experiments

## D.1   Experimental details

**Computational resources.** We used CPU cores to run this set of experiments. In particular, for each trial, we used 2 CPU cores and up to 16 GB of memory. We ran the experiments in parallel using a shared cluster with 10000 CPU cores approximately.

**General experimental setup.** We repeat all experiments 10 times, which correspond to 10 different random initialization of the parameters of the GNNs. In all cases, we choose the model parameters with the best validation performance during training. In order to run the experiments and collect the results, we used the GraphGym framework [32], which includes the data processing and loading of the datasets, as well as the evaluation and collection of the results. We split the datasets in 70 % training, 15 % validation, and 15 % test.

We cross-validate the number of message-passing layers in the network ($2, 3, 4$), as well as the learning rate ($[0.01, 0.005]$). Then, we report the results of the best validation error among the 4 possible combinations. However, in practice we found the best performance always to use 4 message-passing layers, and thus the only difference in configuration lies in the learning rate.

We use residual connections between the GNN layers, 4 heads in the attention models, and the Parametric ReLU (PReLU) [12] as the nonlinear activation function. We do not use batch normalization [14], nor dropout [24]. We use the Adam optimizer [16] with $\beta = (0.9, 0.999)$, and an exponential learning-rate scheduler with $\gamma = 0.998$. We train all the models for 2500 epochs. Importantly, we do not use weight decay, since this will bias the solution towards $\lambda_1 = 0$ and $\lambda_2 = 1$.

We use the Pytorch Geometric [9] implementation of L-CAT for all experiments, switching between models by properly by setting $\lambda_1$ and $\lambda_2$. We parametrize $\lambda_1$ and $\lambda_2$ as free-parameters in log-space that pass through a sigmoid function—i.e., `sigmoid(`$10^x$`)`—such that they are constrained to the unit interval, and they are learned quickly.

### D.2   Additional results

Table 4 shows the results presented in the main paper (with the addition of a dense feed-forward network), while Table 5 presents the results for the remaining datasets, with smaller average degree.

If we focus on Table 5, we observe that all models perform equally well, yet in a few cases CAT and L-CAT are significantly better than the baselines—e.g., L-CATv2 in *CoauthorCS*, or L-CAT in *Cora*. Following a similar discussion as the one presented in the main paper, these results indicates that L-CAT achieves similar or better performance than baseline models and thus, should be the preferred architecture.

**Competitive performance without the graph.** We also include in Tables 4 and 5 the performance of a feed-forward network, referred to as Dense (first row). Note that the only data available to this model are the node features, and thus no graph information is provided. Therefore, we should expect a significant drop in performance, which indeed happens for some datasets such as *Amazon Computers* ($\approx 7\%$ drop), *FacebookPagePage* ($\approx 20\%$ drop), *DBLP* ($\approx 9\%$ drop) and *Cora* ($\approx 14\%$ drop). Still, we found that for other commonly used datasets the performance is similar, e.g., *Coauthor Physics* and *PubMed*; or *it is even better CoauthorCS*. These results manifest the importance of a proper benchmarking, and of carefully considering the datasets used to evaluate GNN models.

Table 4: Test accuracy (%) of the considered convolution and attention models for different datasets (sorted by their average node degree), and averaged over ten runs. Bold numbers are statistically different to their baseline model ($\alpha = 0.05$). Best average performance is underlined.

| Dataset | Amazon Computers | Amazon Photo | GitHub | Facebook PagePage | Coauthor Physics | TwitchEN |
|---|---|---|---|---|---|---|
| Avg. Deg. | 35.76 | 31.13 | 15.33 | 15.22 | 14.38 | 10.91 |
| Dense | $83.73 \pm 0.34$ | $91.74 \pm 0.46$ | $81.21 \pm 0.30$ | $75.89 \pm 0.66$ | $95.41 \pm 0.14$ | $56.26 \pm 1.74$ |
| GCN | $\underline{90.59 \pm 0.36}$ | $\underline{95.13 \pm 0.57}$ | $84.13 \pm 0.44$ | $94.76 \pm 0.19$ | $96.36 \pm 0.10$ | $57.83 \pm 1.13$ |
| GAT | $89.59 \pm 0.61$ | $94.02 \pm 0.66$ | $83.31 \pm 0.18$ | $94.16 \pm 0.48$ | $96.36 \pm 0.10$ | $57.59 \pm 1.20$ |
| CAT | $\mathbf{90.58 \pm 0.40}$ | $\mathbf{94.77 \pm 0.47}$ | $\mathbf{84.11 \pm 0.66}$ | $\mathbf{94.71 \pm 0.30}$ | $\underline{96.40 \pm 0.10}$ | $\underline{58.09 \pm 1.61}$ |
| L-CAT | $\mathbf{90.34 \pm 0.47}$ | $\mathbf{94.93 \pm 0.37}$ | $84.05 \pm 0.70$ | $\underline{\mathbf{94.81 \pm 0.25}}$ | $96.35 \pm 0.10$ | $57.88 \pm 2.07$ |
| GATv2 | $89.49 \pm 0.53$ | $93.47 \pm 0.62$ | $82.92 \pm 0.45$ | $93.44 \pm 0.30$ | $96.24 \pm 0.19$ | $57.70 \pm 1.17$ |
| CATv2 | $\mathbf{90.44 \pm 0.46}$ | $\mathbf{94.81 \pm 0.55}$ | $\mathbf{84.10 \pm 0.88}$ | $\mathbf{94.27 \pm 0.31}$ | $96.34 \pm 0.12$ | $57.99 \pm 2.02$ |
| L-CATv2 | $90.33 \pm 0.44$ | $94.79 \pm 0.61$ | $\underline{\mathbf{84.31 \pm 0.59}}$ | $94.44 \pm 0.39$ | $96.29 \pm 0.13$ | $57.89 \pm 1.53$ |

## E   Open Graph Benchmark experiments

### E.1   Experimental details

**Computational resources.** For this set of experiments, we had at our disposal a set of 16 Tesla V100-SXM GPUs with 160 CPU cores, shared among the rest of the department.

**Statistical significance.** For each CAT and L-CAT model, we highlight significant improvements according to a two-sided paired t-test ($\alpha = 0.05$), with respect to its corresponding baseline model. For example, for L-CATv2 with 8 heads we perform the test with respect to GATv2 with 8 heads.

**General experimental setup.** We repeat all experiments with OGB datasets 5 times. In all cases, we choose the model parameters with the best validation performance during training. Moreover,

Table 5: Test accuracy (%) of the considered convolution and attention models for different datasets (sorted by their average node degree), and averaged over ten runs. Bold numbers are statistically different to their baseline model ($\alpha = 0.05$). Best average performance is underlined.

| Model | CoauthorCS | DBLP | PubMed | Cora | CiteSeer |
|---|---|---|---|---|---|
| Avg. Deg. | 8.93 | 5.97 | 4.5 | 3.9 | 2.74 |
| Dense | $\underline{94.88 \pm 0.21}$ | $75.46 \pm 0.27$ | $88.13 \pm 0.33$ | $72.75 \pm 1.72$ | $73.02 \pm 1.01$ |
| GCN | $93.85 \pm 0.23$ | $84.18 \pm 0.40$ | $88.50 \pm 0.18$ | $\underline{86.68 \pm 0.78}$ | $\underline{75.76 \pm 1.09}$ |
| GAT | $93.80 \pm 0.38$ | $84.15 \pm 0.39$ | $\underline{88.62 \pm 0.18}$ | $85.95 \pm 0.95$ | $75.40 \pm 1.43$ |
| CAT | $93.70 \pm 0.31$ | $84.10 \pm 0.29$ | $88.58 \pm 0.25$ | $85.85 \pm 0.79$ | $75.64 \pm 0.91$ |
| L-CAT | $93.65 \pm 0.23$ | $84.13 \pm 0.26$ | $88.45 \pm 0.32$ | $\mathbf{86.66 \pm 0.87}$ | $75.04 \pm 1.12$ |
| GATv2 | $93.19 \pm 0.64$ | $\underline{84.33 \pm 0.18}$ | $88.52 \pm 0.27$ | $85.65 \pm 1.01$ | $75.14 \pm 1.20$ |
| CATv2 | $93.51 \pm 0.34$ | $\mathbf{84.15 \pm 0.41}$ | $88.54 \pm 0.29$ | $85.50 \pm 0.94$ | $74.68 \pm 1.30$ |
| L-CATv2 | $\mathbf{93.65 \pm 0.20}$ | $84.31 \pm 0.31$ | $88.48 \pm 0.24$ | $85.75 \pm 0.72$ | $75.04 \pm 1.30$ |

when we show the results without specifying the number of heads, we take the model with the best validation error among the two models with 1 and 8 heads.

We use the same implementation of L-CAT for all experiments, switching between models by properly setting $\lambda_1$ and $\lambda_2$. Experiments on *arxiv*, *mag*, *products* use a version of L-CAT implemented in Pytorch Geometric [9]. Experiments on *proteins* use a version of L-CAT implemented in DGL [28]. We parametrize $\lambda_1$ and $\lambda_2$ as free-parameters in log-space that pass through a sigmoid function—i.e., `sigmoid(10^x)`—such that they are constrained to the unit interval, and they are learned quickly.

**ArXiv.** We use the example code from the OGB framework [13]. The network is composed of 3 GNN layers with a hidden size of 128. We use batch normalization [14] and a dropout [24] of 0.5 between the GNN layers, and Adam [16] with a learning rate of 0.01. We use the ReLU as activation function. For the initial experiments, we train for 1500 epochs, while we train for 500 epochs for the noise experiments.

**MAG.** We adapted the official code from [7]. The network is composed of 2 layers with 128 hidden channels. This time, we use layer normalization [2] and a dropout of 0.5 between the layers. Again, we use ReLU as the activation function, and add residual connections to the network. As with *arxiv*, we use Adam [16] with learning rate 0.01. We set a batch size of 20000 and train for 100 epochs.

**Products.** We use the same setup as [7], with a network of 3 GNN layers and 128 hidden dimensions. We apply residual connections once again, with a dropout [24] of 0.5 between layers. This time, we use ELU as the activation function. The batch size is set to 256. Adam [16] is again the optimizer in use, this time with a learning rate of 0.001. We train for 100 epochs, although we apply early stopping whenever the validation accuracy stops increasing for more than 10 epochs. Note the training split of this dataset only contains 8 % of the data.

**Proteins.** We follow once more the setup of [7]. The network we use has 6 GNN layers of hidden size 64. Dropout [24] is set to 0.25 between layers, with an input dropout of 0.1. At the beginning of the network, we place a linear layer followed by a ReLU activation to encode the nodes, and a linear layer at the end of the network to predict the class. Moreover, we use batch normalization [14] between layers and ReLU as the activation function. We train the model for 1200 epochs at most, with early stopping after not improving for 10 epochs.

### E.2 Additional results

We show in Tables 6 to 8 the results for the *arxiv*, *mag*, and *products* datasets, respectively, without selecting the best configuration for each type of model. That is, we show the results for both number of heads.

**Extrapolation ablation study.** Due to page constraints, these results were not added to the main paper. Here, we study two questions. First, how important are $\lambda_1$ and $\lambda_2$ in the formulation of L-CAT (Eq. 5)? For the sake of completeness, the second question we attempt to answer here is whether we can obtain similar performance by just interpolating between GCN and GAT (fixing $\lambda_2 = 0$)? Note

Table 6: Test accuracy on the *arxiv* dataset for attention models using 1 head and 8 heads.

|      | GCN             | GAT             | CAT                    | L-CAT               | GATv2           | CATv2                  | L-CATv2         |
| ---- | --------------- | --------------- | ---------------------- | ------------------- | --------------- | ---------------------- | --------------- |
| 1h   | $71.58 \pm 0.19$ | $71.58 \pm 0.15$ | **$\underline{72.04 \pm 0.20}$** | **$72.00 \pm 0.11$** | $71.70 \pm 0.14$ | **$72.02 \pm 0.08$** | $71.96 \pm 0.21$ |
| 8h   | –               | $71.63 \pm 0.11$ | **$\underline{72.14 \pm 0.20}$** | **$71.98 \pm 0.08$** | $71.72 \pm 0.24$ | $71.76 \pm 0.14$ | $71.91 \pm 0.16$ |

Table 7: Test accuracy on the *mag* dataset for attention models using 1 head and 8 heads.

|      | GCN                       | GAT             | CAT                 | L-CAT               | GATv2                    | CATv2                  | L-CATv2         |
| ---- | ------------------------- | --------------- | ------------------- | ------------------- | ------------------------ | ---------------------- | --------------- |
| 1h   | $\underline{32.77 \pm 0.36}$ | $32.35 \pm 0.24$ | $31.98 \pm 0.46$    | $32.47 \pm 0.38$    | $32.76 \pm 0.18$         | **$32.43 \pm 0.22$** | $32.68 \pm 0.50$ |
| 8h   | –                         | $32.15 \pm 0.31$ | **$31.58 \pm 0.22$** | $32.49 \pm 0.21$    | $\underline{32.85 \pm 0.21}$ | **$32.34 \pm 0.18$** | **$32.38 \pm 0.28$** |

Table 8: Test accuracy on the *products* dataset for attention models using 1 head and 8 heads.

|      | GCN             | GAT                       | CAT                 | L-CAT               | GATv2           | CATv2           | L-CATv2             |
| ---- | --------------- | ------------------------- | ------------------- | ------------------- | --------------- | --------------- | ------------------- |
| 1h   | $74.12 \pm 1.20$ | $\underline{78.53 \pm 0.91}$ | **$77.38 \pm 0.36$** | $77.19 \pm 1.11$    | $73.81 \pm 0.39$ | $74.81 \pm 1.12$ | **$76.37 \pm 0.92$** |
| 8h   | –               | $\underline{78.23 \pm 0.25}$ | **$76.63 \pm 1.15$** | **$76.56 \pm 0.45$** | $76.40 \pm 0.71$ | $75.20 \pm 0.92$ | **$74.70 \pm 0.28$** |

that we theoretically showed in §§4 and 6 that CAT fills up a gap between GCN and GAT, making it preferable in certain settings.

We include three additional models: GCN-GAT, which interpolates between GCN and GAT (or GATv2) by learning $\lambda_1$ and fixing $\lambda_2 = 0$; CAT-$\lambda_1$ which interpolates between GCN and CAT by learning $\lambda_1$ and fixing $\lambda_2 = 1$; and CAT-$\lambda_2$, which interpolates between GAT and CAT by learning $\lambda_2$ and fixing $\lambda_1 = 1$.

Results using GAT and shown in Table 9, and using GATv2 in Table 10. We can observe that GCN-GAT obtains results in between GCN and GAT for all settings, despite being able to interpolate between both layers in each of the six layers of the network. Regarding learning $\lambda_1$ and $\lambda_2$, we can observe that there is a clear difference between learning boths (L-CAT), and learning a single one. For both attention models, CAT-$\lambda_1$ obtains better results than CAT-$\lambda_2$ in all settings, but *uniform* with 8 heads. Still, the results of both variants are substantially worse than those of L-CAT in all cases, *demonstrating the importance of learning to interpolate between the three layer types*.

Table 9: Test accuracy on the *proteins* dataset for GCN [17] and GAT [26] attention models using two network initializations, and two numbers of heads (1 and 8).

|      | GCN                       | GCN-GAT             | GAT             | CAT                 | L-CAT                    | CAT-$\lambda_1$      | CAT-$\lambda_2$      |
| ---- | ------------------------- | ------------------- | --------------- | ------------------- | ------------------------ | ------------------- | ------------------- |
|      |                           |                     | *uniform* initialization |          |                          |                     |                     |
| 1h   | $61.08 \pm 2.86$          | **$70.44 \pm 1.56$** | $59.73 \pm 4.04$ | **$74.19 \pm 0.72$** | **$77.77 \pm 1.44$**    | **$71.97 \pm 3.78$** | **$73.55 \pm 1.36$** |
| 8h   | –                         | **$68.51 \pm 0.91$** | $72.23 \pm 3.20$ | $73.60 \pm 1.27$    | $\underline{78.85 \pm 1.76}$ | $76.43 \pm 2.47$    | $72.76 \pm 2.79$    |
|      |                           |                     | *normal* initialization |           |                          |                     |                     |
| 1h   | $\underline{80.10 \pm 0.61}$ | $66.51 \pm 3.23$    | $66.38 \pm 7.76$ | $73.26 \pm 1.84$    | **$78.06 \pm 1.40$**    | $76.77 \pm 1.91$    | $73.39 \pm 1.25$    |
| 8h   | –                         | **$69.93 \pm 1.93$** | $79.08 \pm 1.64$ | **$74.67 \pm 1.29$** | $\underline{79.63 \pm 0.79}$ | $78.86 \pm 1.07$    | **$73.32 \pm 1.15$** |

Table 10: Test accuracy on the *proteins* dataset for GCN [17] and GATv2 [7] attention models using two network initializations, and two numbers of heads (1 and 8).

|      | GCN                       | GCN-GATv2           | GATv2           | CATv2                | L-CATv2                  | CATv2-$\lambda_1$    | CATv2-$\lambda_2$    |
| ---- | ------------------------- | ------------------- | --------------- | -------------------- | ------------------------ | ------------------- | ------------------- |
|      |                           |                     | *uniform* initialization |         |                          |                     |                     |
| 1h   | $61.08 \pm 2.86$          | **$69.69 \pm 1.59$** | $59.85 \pm 3.05$ | **$64.32 \pm 2.61$** | **$79.08 \pm 1.06$**    | $63.24 \pm 1.55$    | **$73.41 \pm 0.34$** |
| 8h   | –                         | **$69.94 \pm 1.62$** | $75.21 \pm 1.80$ | $74.16 \pm 1.45$     | $\underline{78.77 \pm 1.09}$ | $77.61 \pm 1.32$    | $73.96 \pm 1.27$    |
|      |                           |                     | *normal* initialization |          |                          |                     |                     |
| 1h   | $\underline{80.10 \pm 0.61}$ | $68.54 \pm 1.63$    | $69.13 \pm 9.48$ | $74.33 \pm 1.06$     | **$79.07 \pm 1.09$**    | $78.41 \pm 0.93$    | $74.07 \pm 1.17$    |
| 8h   | –                         | **$68.71 \pm 1.96$** | $78.65 \pm 1.61$ | **$73.40 \pm 0.62$** | $\underline{79.30 \pm 0.55}$ | $78.76 \pm 1.41$    | **$73.22 \pm 0.77$** |

# F Future work and societal impact

Given the nature of this work, we do not see any social concerns in this work. On the contrary, L-CAT eases the applicability of GNNs to the practitioner, and removes the need of cross-validating the layer type, which can potentially benefit other areas and applications, as GNNs have already proven.

We strongly believe learnable interpolation can get us a long way, and we hope L-CAT to motivate new and exciting work. For example, it would be interesting to see L-CAT applied to other GCN and GAT variants, such as those in [15, 25, 30]. Specially, we are eager to see L-CAT in real applications, and thus finding out what combining different GNN layers across a model (without the hurdle of cross-validating all combinations) can lead to in the real-world.

