# OpenReview forum: "Learnable Graph Convolutional Attention Networks"
_NeurIPS.cc/2022/Workshop/HITY — HITY Workshop NeurIPS 2022_

### Official Review · Reviewer_WNwn · 2022-10-13

**Rating:** 1
**Confidence:** 3

**Review:**

**Summary:** The authors propose a new architecture (CAT for Convolutional
Attention Layer) for Graph Neural Networks that combines GCNs (Graph
Convolutional Networks) and GATs (Graph Attention Networks). An example with
synthetic data shows (theoretically and empirically) that the optimal choice
among the three architectures depends on the data. Thus, the paper introduces
L-CAT (Learnable Convolutional Attention Layer) which is able to interpolate
between CAT, GCN and GAT.

**Strengths, Weaknesses & Questions:**
- Overall, the paper is well-structured and nicely written. The introduction
clearly explains the context of the work and the contribution of the paper. The
second section *Preliminaries* provides a good overview on GNNs. Theoretical
findings are discussed and also validated empirically.
- Line 63-69: As a non-expert, I have difficulties understanding the CSBM
example. I would have appreciated if you had provided more *intuition* on what
the quantities mean and what their impact on separation *difficulty* is
(especially for the quantities $p$, $q$ and $\Vert\mu\Vert$).
- Line 113-115: You claim that "CAT and L-CAT [...] mostly improve test accuracy
with respect to their baseline model" (I guess this baseline refers to GCN?).
I'm confused about how you arrived at this statement since there is only one
case (*Facebook PagePage*) out of six, where CAT or L-CAT significantly
outperform the GCN baseline. In all other cases, if I understand correctly, GCN
is better on average or the improvements of CAT/L-CAT are insignificant.

**Minor:**
- Line 37: At this point, I was wondering what the objective for the GNN was. Is
it node classification, learning a node embedding, or something else?
- Line 47: What does it mean for a graph to be *noisy*? It seems that this
refers to the parameter $q$ (line 104) but also to $\sigma$ (as in the middle
figure on page 4). Maybe clarify this by providing a more precise definition.
- Theorem 1 and Corollary 2: What are $\Omega$ and $\omega$?
- Table 1: In the *GitHub* column, none of the results are underlined.
- Top figure on page 4: I don't understand what is shown here. What is the
average node degree and what does the accuracy improvement on the y-axis refer
to?

---

### Official Review · Reviewer_J91H · 2022-10-16
**Accept: The paper proposes an interesting approach to improve graph neural networks**

**Rating:** 1
**Confidence:** 3

**Review:**

The paper proposes a new layer type for graph neural networks, which allows to interpolate between a graph convolutional, a graph attention, and a graph convolutional attention layer. This is achieved by introducing two additional scalar parameters per layer, which are jointly learned end-to-end with the other parameters. The authors provide motivating theoretical results based on previous work and empirically validate their proposed method.

Overall, interpolating between layer types with parameters which can be learned end-to-end seems like a promising direction to improve graph neural networks. Hence, I recommend to accept the paper.

---

### Decision · Program_Chairs · 2022-10-20

Accept